# The uncertainty related to the inexactitude of prioritization based on consistent pairwise comparisons

**Pawel Tadeusz Kazibudzki**�ID*

Faculty of Economics and Management, Opole University of Technology, Opole, Poland

* p.kazibudzki@po.edu.pl

**Data Availability Statement:** All relevant data for this study are publicly available from the Zenodo repository (https://doi.org/10.5281/zenodo.8071858).

## Abstract

When the in/consistency in Pairwise Comparisons (PCs) is taken into consideration as the subarea of the Multi Attribute Decision Making (MADM) scientific field, it has many repercussions in various types of research areas including different modelling scenarios e.g. reduction of inconsistency during PCs, deriving appropriate consistency thresholds for inconsistent Pairwise Comparison Matrices (PCMs), completing of incomplete PCMs, aggregating of individual PCMs in relation to Group Decision Making (GDM) aspects, and PCMs in/consistency relation to credibility of Priority Vectors (PV) derived from PCMs with the application of various Priorities Deriving Methods (PDMs). The examination objective in the latter area of research is the uncertainty related to the inexactitude of prioritization based on derived PVs. However, only few research studies examine this problem from the perspective of PCM applicability for credible designation of decision maker's (DM) priorities in the way that leads to minimization of the prioritization uncertainty related to possible, and sometimes very probable, ranking fluctuations. This problem constitutes the primary area of interest for this research paper as no research study was thus far identified that examines this problem from the perspective of consistent PCMs. Hence, a research gap was identified. Thus, the objective of this research paper is to fill in this scientific gap. The research findings have serious repercussions in relation to prioritization quality with the application of PCs methodology, mostly in relation to the interpretation and reliability evaluation of prioritization results. Firstly, the research study outcome changes the perspective of the rank reversal phenomenon, which shed new light on many research studies that have been presented in the subject's literature for many decades. Secondly, the research study results throw new light on the discussion concerning the fuzziness of AHP's results. Last but not least, the effect of the research opens the unique opportunity to evaluate the prioritization outcome obtained within the process of consistent PCs from the well-known perspective of statistical hypothesis testing i.e. the probability designation of the chance that accepted ranking results which were considered as correct due to low probability of change may be incorrect, hence they should be rejected, and the probability designation of the chance that rejected ranking results which were considered as incorrect due to high probability of change may be correct and should be accepted. The paramount finding of the research is the fact that consistent PCMs provide PVs, which elements cannot be considered as

**Funding:** This work was financially supported by the Opole University of Technology under GAMMA project no. 152/22. The APC was partially funded by Opole University of Technology, Poland. The APC funders had no role in the design of the study; in the collection, analyses, or interpretation of data; in the writing of the manuscript, or in the decision to publish the results.

**Competing interests:** The authors have declared that no competing interests exist.

established, but only approximated within certain confidence intervals estimated with a certain level of probability. As problems related to heuristics can be analyzed only via a computer simulation process, because they cannot be mathematically determined, the problem examined in this research paper is examined via Monte Carlo simulations, appropriately coded and executed with the application of Wolfram's Mathematica Software. It is believed that this research findings should be very important and useful for all decision makers and researchers during their problems' examinations that relate to prioritization processes with the application of PCs methodology.

## Introduction

Due to the fact acknowledged by many research studies, including Miller's input, [1] concluding that humans are not capable of dealing accurately with more than about seven (±2) things at a time (the human brain is limited in its short term memory capacity, its discrimination ability and its range of perception), there are numerous techniques/methods which strive to make this process easier and sometimes even possible. Fundamentally, two schools of decision making methodology exist at the present moment: the Multiple Criteria Decision Making (MCDM) school, developed by Americans, and the Multiple Criteria Decision Aiding/Analysis (MCDA) school, developed by Europeans [2–4]. For most researchers, these terms have similar meaning, but for the record, the Americans' school is applied in this research. Terms like Multi-Objective Decision Making (MODM), Multi-Attributes Decision Making (MADM), as well as Multi-Dimensions Decision-Making (MDDM) can also be found in literature. Generally, when optimization techniques are utilized (continuous problems are examined), one deals with MODM, and when alternative selection takes place (discrete problems are considered), one deals with MADM and/or MDDM. The facet of MCDM is created by these subdisciplines all together.

Pairwise-Comparisons-Based (PCB) prioritization is a method with a long history dating back to the Middle Ages. Presumably, the first elaboration on this subject was created by Ramon Lull [5], who in his work debated comparisons of alternatives in an election process. Over time, other studies on the PCs method appeared, such as the Condorcet and the Copeland methods, see e.g. [6–8]. Indisputably, due to Saaty's manuscript [9] where he defined the Analytic Hierarchy Process (AHP), alternatives comparisons in pairs, contemporarily called pairwise comparisons, started to be considered an element of the MADM.

Undeniably, the AHP is a popular MADM method that proposes its own priorities deriving method (PDM) i.e. Principal Right Eigenvector (PREV) method, a related to PREV consistency index (CI) which is supposed to indicate quality of data which is processed and a hierarchical model which is supposed to enable easier structuring of multiple criteria problems [10–13]. Over time, scientific evidence also appeared that indicated a few flaws of the AHP, see e.g. [14–17]. Due to some criticism of the AHP methodology, many scientists worked and keep working on methods which can improve its drawbacks. Hence, many PDMs have been proposed, see e.g. [18–50], which, for the article's brevity, will not be discussed herein in details.

Taking into account the AHP drawbacks, many indicators of PCM consistency,–commonly known as consistency indices (CIs)–have been also proposed thus far, see e.g. [23,24,41,51–68]. They also, due to brevity of this article, will not be scrutinized herein. However, the interested reader may choose to acquaint thoroughly with their ample diversity, see e.g. [69–77].

When the in/consistency in Pairwise Comparisons (PCs) is taken into consideration as the subarea of the MADM scientific field, it presumably may be perceived as the most exploiting topic in this research area. A variety of models have been proposed to address inconsistency

issues, see e.g. [78–85]. Certainly, issues related to PCs in/consistency have many repercussions in various types of modelling scenarios i.e. the inconsistency reduction of reciprocal Pairwise Comparison Matrices (PCMs) with high levels of inconsistency, see e.g. [72,86–91]; deriving appropriate consistency thresholds for non/reciprocal PCMs, see e.g. [10,74,92–96]; completing of incomplete reciprocal PCMs, see e.g. [97–102]; and aggregating of individual reciprocal PCMs in relation to Group Decision Making (GDM) aspects, see e.g. [75,103–110]. The issue of in/consistency in PCs is also especially attractive when examined from the perspective of its relation to trustworthiness of Priority Ratios (PRs) derived from Pairwise Comparison Matrix (PCM) denoted as $PCM(w) = [w_{ij}]_{n \times n}$ with elements $w_{ij} = w_i/w_j$, where $w_{ij} > 0$, and $i, j = 1, \ldots, n$, with the application of various Priorities Deriving Methods (PDMs). It is believed for example, although some evidence from a few research papers contradicts this belief, see e.g. [76,77,111,112], that inconsistent PCMs provide less credible PRs and inconsistency reduction is the process that leads to betterment of priorities estimation. It is also believed that both consistent and inconsistent PCMs provide Priority Vectors (PVs), denoted as $w = [w_1, \ldots, w_n]^T$, where $w_i > 0$, $i = 1, \ldots, n$, whose elements are considered as established, and not approximated within a certain confidence intervals estimated with a certain level of probability. The PCMs in/consistency relation to credibility of Priority Vectors (PV) derived from PCMs with the application of various Priorities Deriving Methods (PDMs) constitutes the key issue in a few research studies e.g. [51,76,77,111–117]. The examination objective in the latter area of research is the uncertainty related to the inexactitude of prioritization based on derived PVs. However, only few research studies examine this problem from the perspective of PCM applicability for credible designation of decision maker's (DM) priorities in the way that leads to minimization of the prioritization uncertainty related to possible, and sometimes very probable, ranking fluctuations. This problem constitutes the primary area of interest for this research paper as no research study was thus far identified that examines this problem from the perspective of consistent PCMs. So far, this concept has been studied only from the perspective of inconsistent PCMs, see e.g. [51,76,77,111,112,114,117]. Hence, a research gap was identified. Thus, the objective of this research paper is to fill in this scientific gap. The research findings have serious repercussions in relation to prioritization quality with the application of PCs methodology, mostly in relation to the interpretation and reliability evaluation of prioritization results. Firstly, the research study outcome changes the perspective of the rank reversal phenomenon, which shed new light on many research studies that have been presented in the subject's literature for many decades. Secondly, the research study results throw new light on the discussion concerning the fuzziness of AHP's results. Last but not least, the effect of the research opens the unique opportunity to evaluate the prioritization outcome obtained within the process of consistent PCs from the well-known perspective of statistical hypothesis testing i.e. the probability designation of the chance that accepted ranking results which were considered as correct due to low probability of change may be incorrect, hence they should be rejected, and the probability designation of the chance that rejected ranking results which were considered as incorrect due to high probability of change may be correct and should be accepted. The paramount finding of the research is the fact that consistent PCMs provide PVs, which elements cannot be considered as established, but only approximated within certain confidence intervals estimated with a certain level of probability. As problems related to heuristics can be analyzed only via a computer simulation process, because they cannot be mathematically determined, the problem examined in this research paper is examined via Monte Carlo simulations, appropriately coded and executed with the application of Wolfram's Mathematica Software. It is believed that this research findings should be very important and useful for all decision makers and researchers during their problems' examinations that relate to prioritization processes with the application of PCs methodology.

Due to its main area of interest, this research paper is structured as follows: firstly, in the Section *Systematic review of literature*, the MCDM techniques/methods are reviewed, and then some *Preliminary remarks* are presented; secondly, in the Section *Research methodology*, the research method of the problem is presented; this Section consists of two Subsections i.e. *Outline of the examination concept*, where the example case study is analyzed, and *Description of simulation concept*, where the Monte Carlo simulation algorithm applied for the research is presented; thirdly, the Section *Results and discussion*, which is also divided into two Subsections i.e. *Analysis of results*, where general simulation results are elaborated, and *Discussion of results*, where detailed simulation results are examined and discussed from the perspective of Rank Reversal Phenomenon; fourthly, final remarks and future research direction end the research paper with the Section *Final remarks*.

## Systematic review of literature

There are many MCDM techniques/methods which are available in relevant literature. As they have their own unique characteristics, there are many ways to classify them e.g. according to the type of data they utilize, according to the number of Decision Makers (DM) involved in the decision process, or according to the type of information and pertinent features of given information.

Thus far, the following techniques/methods have been devised for MADM/MDDM problems: the Weighted Sum Model (WSM), see e.g. [118,119], useful for evaluating several alternatives in relation to various criteria expressed in the same unit; the Weighted Product Model (WPM), see e.g. [120], often called dimensionless analysis because its mathematical structure eliminates any units of measure (the WPM can be applied to both, single- and multi-dimensional MCDM problems); the Analytic Hierarchy Process (AHP), see e.g. [17,37,42,69,121–129], which will be examined herein in more details from the perspective of this research paper's objective; the Analytic Network Process (ANP), see e.g. [130–135], which expands the AHP concept for situations with dependence and feedback among alternatives and criteria; the fuzzy AHP (F-AHP) which implements the concepts of Zadeh's [136] fuzzy set theory, see e.g. [11,137–142]; the Data Envelopment Analysis (DEA), see e.g. [143–145], which is used to estimate an efficiency frontier by considering the best performance observations (extreme points) which 'envelop' the remaining observations; Goal Programming (GP), see e.g. [20,146–150], which is used for solving multi-objective optimization problems that balance a trade-off in conflicting objectives; Grey Analysis (GA), see e.g. [119,151–153] which applies a sophisticated mathematical analysis of the systems which are partly defined and partly unknown, thus recognized as 'insufficient data' and 'weak knowledge'; ELECTRE (in French: ELimination Et Choix Traduisant la REalité), see e.g. [3,154–156], implemented to select the best alternative with maximum advantage and least conflict in relation to various criteria; VIKOR (from the Serbian: Vise Kriterijumska Optimizacija I Kompromisno Resenje), see e.g. [157,158], applied for examination of alternative preferences in highly complex environments; Technique for Order Preference by Similarity to Ideal Solution (TOPSIS), see e.g. [158–164], which finds the best solutions of MADM problems looking for the shortest Euclidean distance from the positive-ideal solution, and the longest Euclidean distance from the negative-ideal one; Decision Making Trial and Evaluation Laboratory (DEMATEL), see e.g. [165–172], which deals with examination of interdependent relationships among analyzed factors and identification of the critical ones through a visual structural model; and Measuring Attractiveness by a Categorical Based Evaluation *Technique (*MACBETH), see e.g. [173–175], which is the MADM method that evaluates options against multiple criteria.

When the AHP is examined as the MADM method, contradiction between its easy to use applicability and controversial methodology which it uses is clearly perceived for a long time.

Perhaps, it is due to a few areas that are very intriguing when this methodology is considered i.e. the inconsistency associated with pairwise comparisons made by humans during its methodological process, see e.g. [70–72,78,79,114,176–179]; human judgments' errors connected with human's preferences expression, see e.g. [73,103–105,179–184]; limitations in relation to a selected preference scale which must be applied during the pairwise comparisons process, see e.g. [27,117,185–189]; the reciprocity condition imposed for a Pairwise Comparison Matrix (PCM)–denoted as $PCM(w) = [w_{ij}]_{n\times n}$ with elements $w_{ij} = w_i/w_j$, where $w_{ij}>0$, and $i, j = 1,\ldots, n$, which constitutes the information source for a prioritization process of DM preferences, see e.g. [39,42,46,73,78,184,190]; and last but not least, the Condition of the order Preservation (COP), which constitutes a complementary approach to the examination of the in/consistency of PCM, see e.g. [15,32,191–195].

When the in/consistency in Pairwise Comparisons (PCs) is taken into consideration as the subarea of the Multi Attribute Decision Making (MADM) scientific field, it has many repercussions in various types of research areas including different modelling scenarios e.g. reduction of inconsistency during PCs, deriving appropriate consistency thresholds for inconsistent Pairwise Comparison Matrices (PCMs), completing of incomplete PCMs, aggregating of individual PCMs in relation to Group Decision Making (GDM) aspects, and PCMs in/consistency relation to credibility of Priority Vectors (PV) derived from PCMs with the application of various Priorities Deriving Methods (PDMs). The examination objective in the latter area of research is the uncertainty related to the inexactitude of prioritization based on derived PVs.

It should be realized here that there are three significantly different notions:

- the PCM consistency perceived from the perspective of its definition, see hereafter D[3], and expressed by the specific inconsistency index value;

- the consistency of decision makers, i.e. their trustworthiness, reflected by the number and size of their judgments discrepancies, and;

- the PCM applicability for estimation of decision makers' priorities in the way that leads to minimization of their estimation errors.

As it seems the third issue is probably the most important problem in the contemporary arena of the MADM theory concerning AHP, and the only way to examine that phenomena is through computer simulations. It is the fact that Monte Carlo simulations are commonly recognized and applied as important and credible source of scientific information [196,197]. Their applications spread for examination purposes of various phenomena, e.g.: consequences of decisions made, or different processes subdued to random impact of the particular environment [32,49,76,113,117,198–200].

## Preliminary remarks

It is paramount to underline, that all of so far devised PDMs provide exactly the same results i.e. priorities ratios (PR) within priority vectors (PV) when consistent PCMs are processed as the source of data. In relation to the subject of this research paper, it is paramount to emphasize at this point, that for perfectly consistent PCMs, all those thus far devised and potentially yet not invented indicators of PCM consistency must be by definition equal to zero. Hence, for the problem examined in this paper, it does not matter which priorities deriving method (PDM), and which consistency index (CI), out of all available in literature, will be utilized hereafter.

Taking into account possible distortions of priorities ratios (PRs) derived from consistent PCMs, the three factors, out of the few listed earlier, must be taken into consideration during

these distortion examinations i.e. human judgment errors connected with human preferences expression, rounding errors resulting from a selected preference scale which must be applied during the pairwise comparisons process, and the PCM reciprocity requirement commonly imposed on this source of computational data. All of them can be examined with the application of Monte Carlo simulations applied for the purpose of this research paper. However, before the introduction of particulars, a few notions must be first presented. Thus, the following definitions D[1–3] are introduced as follows:

**D[1]**: If the elements of a matrix $W(w)$ satisfy the condition $w_{ij} = 1/w_{ji}$ for all $i, j = 1,\ldots, n$, then the matrix $W(w)$ is called *reciprocal*.

**D[2]**: If the following conditions are true: (a) if for any $i = 1,\ldots, n$, an element $w_{ij}$ is not less than an element $w_{ik}$, then $w_{ij} \geq w_{ik}$ for $i = 1,\ldots, n$, and (b) if for any $i = 1,\ldots, n$, an element $w_{ji}$ is not less than an element $w_{ki}$, then $w_{ji} \geq w_{ki}$ for $i = 1,\ldots, n$, then the matrix $W(w)$ is called an *ordinal transitive*.

**D[3]**: If the elements of a matrix $W(w)$ satisfy the condition $w_{ik}w_{kj} = w_{ij}$ for all $i, j, k = 1,\ldots, n$, and the matrix $W(w)$ is *reciprocal*, then it is called *consistent* or a *cardinal transitive*.

While the norm of the vector $w$ can be written as:

$$\|w\| = e^T w \tag{1}$$

where $e = [1, 1,\ldots, 1]^T$, the vector $w$ can be normalized by dividing it by its norm. Thus, for uniqueness, $w$ is referred hereafter in its normalized form.

For every given priority vector $v = [v_1, v_2, v_3,\ldots,v_n]^T$, the Matrix of Priority Ratios (MPR) can be denoted as:

$$MPR(v) = [v_{ij}]_{n \times n} \tag{2}$$

for all $i, j = 1,\ldots, n$.

Reconsidering rounding errors resulting from a given preference scale which can be selected for the pairwise comparisons process, the following integer based preference scales have been proposed thus far, whose numbers are combined with linguistic variables expressing the preference intensity from 'indifferent' to 'extremely preferred' (Tables 1 and 2).

Last but not least, reconsidering human judgment errors connected with human preferences expression, the following relation has been proposed to make their simulation process possible (Formula 1):

$$w_{ij} = \varepsilon_{ij} v_{ij} \tag{3}$$

where $\varepsilon_{ij}$ acts as a perturbation factor oscillating close to unity i.e. it is randomly drawn from an assigned interval e.g. $\varepsilon_{ij} \in [0.8, 1.2]$, and $w_{ij}$ and $v_{ij}$ are the elements of the matrices $PCM(w) = [w_{ij}]_{n \times n}$ and $MPR(v) = [v_{ij}]_{n \times n}$, respectively. In the statistical approach, $\varepsilon_{ij}$ reflects the realization of a random variable which is applied during a simulation process with a certain probability distribution (PD) reflecting imperfect human judgments during pairwise comparisons. In literature, the following types of PDs are commonly taken for similar implementation purposes: *truncated-normal*, *gamma*, *log-normal*, and *uniform* [38,49]. However, in addition to the above listed most utilized types of PDs, one can also find applications of *triangular*, *beta*, Cauchy, and Laplace PDs [26], as well as Fisher–Snedecor PDs which have been recently applied for the first time by Kazibudzki [32]. The maximal possible spread for $\varepsilon_{ij}$ encountered during similar research studies has never been beyond the following interval $\varepsilon_{ij} \in [0.01, 1.99]$.

## Research methodology

### Outline of the examination concept

Let the exemplary PV be considered as $v = [0.35, 0.19, 0.16, 0.30]^T$, which reflects some real, not derived, priorities ratios (PRs) towards some known four objects whose relative characteristics are known i.e. can be computed on the basis of their mass, volume, circumference, etc. Then, the $MPR(v) = [v_{ij}]_{4\times4}$ can be constructed as follows:

$$MPR(v) = \begin{bmatrix} 1 & 1.8421 & 2.1875 & 1.1667 \\ 0.5429 & 1 & 1.1875 & 0.6333 \\ 0.4571 & 0.8421 & 1 & 0.5333 \\ 0.8571 & 1.5789 & 1.8750 & 1 \end{bmatrix} \tag{4}$$

Let the elements of the above $MPR(v)$ be distorted via a perturbation factor of $\varepsilon = 0.9$. As this process leads to reflection of imperfect human judgments during pairwise comparisons, let the new matrix obtained from $MPR(v)$ be denoted as $PCM(w) = [w_{ij}]_{4\times4}$, and be presented as follows:

$$PCM(w) = \begin{bmatrix} 1 & 1.6579 & 1.9688 & 1.0500 \\ 0.4886 & 1 & 1.0688 & 0.5700 \\ 0.4114 & 0.7579 & 1 & 0.4800 \\ 0.7714 & 1.4211 & 1.6875 & 1 \end{bmatrix} \tag{5}$$

Let the above $PCM(w)$ be made reciprocal now, denoted as $PCM_R(w)$, and presented as follows:

$$PCM_R(w) = \begin{bmatrix} 1 & 1.6579 & 1.9688 & 1.0500 \\ 0.6032 & 1 & 1.0688 & 0.5700 \\ 0.5079 & 0.9357 & 1 & 0.4800 \\ 0.9524 & 1.7544 & 2.0833 & 1 \end{bmatrix} \tag{6}$$

**Table 1. The principal preference scale proposed by Saaty [129,201].**

| Intensity of importance | Definition |
|---|---|
| 1 | Equal importance |
| 3 | Weak importance of one over the other |
| 5 | Essential or strong importance |
| 7 | Demonstrated or very strong importance |
| 9 | Absolute or extreme importance |
| 2, 4, 6, 8 | Intermediate values between the two adjacent judgments |
| Reciprocals of the above | If activity $i$ has one of the above nonzero numbers assigned to it when compared with activity $j$, then $j$ has the reciprocal value when compared with $i$. |

**Table 2. Various preference scales devised for pairwise comparisons of alternatives.**

| Scale type | Definition* | Parameters | Comment |
|---|---|---|---|
| Linear | $c = a \cdot x$ | $a > 0$; $x = \{1,2,\ldots,9\}$ | [9] |
| Power | $c = x^a$ | $a > 1$; $x = \{1,2,\ldots,9\}$ | [202] |
| Geometric | $c = a^{x-1}$ | $a > 1$; $x = \{1,2,\ldots,9\}$ or $x = \{1,1.5,\ldots,4\}$ or other step | [203,204] |
| Logarithmic | $c = \log_a(x+(a-1))$ | $a > 1$; $x = \{1,2,\ldots,9\}$ | [205] |
| Root | $c = \sqrt[a]{xa}$ | $a > 1$; $x = \{1,2,\ldots,9\}$ | [202] |
| Asymptotical | $c = \tanh^{-1}\left(\frac{\sqrt{3}(x-1)}{14}\right)$ | $x = \{1,2,\ldots,9\}$ | [206] |
| Inverse Linear | $c = 9/(10-x)$ | $x = \{1,2,\ldots,9\}$ | [186] |
| Balanced | $c = w/(1-w)$ | $w = \{0.5,0.55,0.6,\ldots,0.9\}$ | [66,207] |
| Balanced Power | $c = 9^{\frac{x-1}{n-1}}$ | $x \in \{1,2,\ldots,n\}$ | [208] |
| Generalized Balanced | $c = \frac{9+(n-1)x}{9+n-x}$ | $x = \{1,2,\ldots,9\}$, $n$–number of criteria | [209] |
| Adaptive | $c = x^{1+\frac{\ln(n-1)}{\ln 9}}$ | $x = \{1,2,\ldots,9\}$, $n$–number of criteria | [209] |
| Adaptive-Balanced | $c = \frac{(9n-10)(x-1)+80}{(9n-10)x-89n+90}(n-1)$ | $x = \{1,2,\ldots,9\}$, $n$–number of criteria | [209] |

\* For the comparison of A and B, c = 1 indicates A = B; c > 1 indicates A > B; when A < B, the reciprocal values 1/c are used.

Last but not least, let the matrix $PCM_R(w)$ be scaled in relation to the linear preference scale proposed by Saaty (Table 1), then denoted as $PCM_{SR}(w)$, and presented as follows:

$$PCM_{SR}(w) = \begin{bmatrix} 1 & 2 & 2 & 1 \\ 0.5 & 1 & 1 & 0.5 \\ 0.5 & 1 & 1 & 0.5 \\ 1 & 2 & 2 & 1 \end{bmatrix} \tag{7}$$

As can be noticed, the matrix $PCM_{SR}(w)$ is perfectly consistent. Hence, any PDM applied herein will derive exactly the same PV which can be denoted as $w$ and presented as follows: $w = [0.333(3), 0.166(6), 0.166(6), 0.333(3)]^T$.

In this presentation, it becomes possible to compare rankings provided by the initial real $v$, and $w$ derived from $PCM_{SR}(w)$ which represents–in the provided example–the possible outcome of DM efforts during the process of $MPR(v)$ approximation. It is also possible to assess the range of deviation between $v$, and $w$. For this purpose, proposed is the use the following two measures i.e. the commonly applied Absolute Average Error (*AAE*) between $v$, and $w$, as defined in Formula 6, and the new, proposed herein, Maximal Absolute Deviation (*MAD*) from *AAE*, as defined in Formula 7.

$$AAE = \frac{1}{n}\sum_{i=1}^{n}|w_i - v_i| \tag{8}$$

$$MAD = max||w_i - v_i| - \frac{1}{n}\sum_{i=1}^{n}|w_i - v_i||_{i=1,2,\ldots,n} \tag{9}$$

It behooves mentioning that the sum of the both defined above measures i.e. *AAE* and *MAD*, provides information about the Maximal Possible Error (*MPE*) that can occur between the examined PRs of vectors $v$, and $w$. For the above example; *MPE* = 0.033(3), with *AAE* = 0.02, and *MAD* = 0.013(3).

Mentionable, in the presented example, the preferences order provided by the two analyzed PVs is different and can be defined respectively as $v_1 \succ v_4 \succ v_2 \succ v_3$ for $v$, and $w_1 \equiv w_4 \succ w_2 \equiv w_3$ for $w$. The latter result clearly indicates that the derived preferences are not as sharp as true ones, and they are definitely more ambiguous in their interpretation. Thus, there is always a margin of error which clearly should be taken into consideration, and thus far, as indicated by all available resources in this research subject, IT IS NOT.

## Description of the simulation concept

The PCMs in/consistency relation to credibility of Priority Vectors (PV) derived from PCMs with the application of various Priorities Deriving Methods (PDMs) constitutes the key issue in a few research studies e.g. [51,76,77,111–117]. The examination objective in this area of research is the uncertainty related to the inexactitude of prioritization based on derived PVs. However, only few research studies examine this problem from the perspective of PCM applicability for credible designation of decision maker's (DM) priorities in the way that leads to minimization of the prioritization uncertainty related to possible, and sometimes very probable, ranking fluctuations. This problem constitutes the primary area of interest for this research paper as no research study was thus far identified that examines this problem via complex simulations from the perspective of consistent PCMs. So far, this concept has been studied only from the perspective of inconsistent PCMs, see e.g. [51,76,77,111,112,114,117]. As problems related to heuristics can be analyzed only via a computer simulation process, because they cannot be mathematically determined, the problem examined in this research paper is examined via Monte Carlo simulations, appropriately coded and executed with the application of Wolfram's Mathematica Software.

It is the fact that Monte Carlo simulations are commonly recognized and applied as important and credible source of scientific information [196,197]. Their applications spread for examination purposes of various phenomena, e.g.: consequences of decisions made, or different processes subdued to random impact of the particular environment [32,49,76,113,117,198–200].

Hence, for the analysis of the problem revealed in the former subsections, the following Monte Carlo simulation algorithm is proposed (Chart 1), and depicted in the form of the Flowchart (Fig 1). The primary version of the algorithm was devised and successfully applied for the first time by Grzybowski [77], and since then it was adapted and successfully implemented for many similar problems, see e.g. [51,76,111,112,114–117,210]. This research study applies the adaptation of this algorithm for examination of consistent PCMs. Three elements in this algorithm are available that have to be defined i.e. the kind of PDM, CI, and the type of preference scale used during its execution as was already stated in earlier subsections of this paper, any kind of PDM and CI suits assumptions of this simulation algorithm. Thus, for simplicity of calculation, the Logarithmic Least Squares Method (LLSM), (appreciated also for its closed form (Formula 8), known as the Geometric Mean Method (GM)), was selected as the PDM [23,24]. It was also decided to use as the simulation algorithm, the modified version of a recently introduced PCM consistency measure [51] i.e. the Index of Absolute Logarithm Deviations (IALD) that is defined herein by Formula 9.

$$w_i = \frac{\left(\prod_{j=1}^{n} w_{ij}\right)^{\frac{1}{n}}}{\sum_{i=1}^{n} \left(\prod_{j=1}^{n} w_{ij}\right)^{\frac{1}{n}}} \tag{10}$$

$$IALD = Median \left| \ln \sum_{j=1}^{n} (w_{ij} w_j / n w_i) \right|_{i=1,\dots,n} \tag{11}$$

As some micro deviations from zero of the applied CI must have appeared during the

simulation research program, it was decided that in order for the obtained results reliability confirmation to apply concurrently as a point of reference, the well-established Koczkodaj's index of consistency applies i.e. $K(A)$ defined herein by the Formula 10.

$$K(A) = \max_{i<j<k} \left\{ TI_p \left( \min \left\{ |1 - \frac{w_{ik}}{w_{ij}w_{jk}}|, |1 - \frac{w_{ij}w_{jk}}{w_{ik}}| \right\} \right) \right\}_{p=1,\dots,} \binom{n}{3} \tag{12}$$

Last but not least, it was decided to also apply three of the available preference scales which were selected on the basis of their features related to pairwise comparison processes i.e. the frequently examined Geometric Scale (GS), in this examination as $a = \sqrt{2}$; $x = \{1,2,\dots,9\}$–linear scale proposed by Saaty for the AHP, and the recently praised Inverse Linear Scale (ILS), see e.g. [185].

## Results and discussion

### Analysis of results

Pursuing the objective of this research, the following general simulation results can be presented. They encompass two sets of possible values for $n$ i.e. $n\in\{3,4,5,6\}$ and $n\in\{4,5\dots,9\}$, and concern two kinds of earlier described errors–$AAE$ and $MAD$. The following two tables–Tables 3 & 4 –contain results for tens of thousands of asymptotically consistent PCMs obtained during millions of iterations of variously distorted PCMs as described in the simulation algorithm presented within Chart 1. In particular, results presented in Table 3 are obtained out of 3,600,000 iterations i.e. 500 various PVs of 4 sizes i.e. $n\in\{3,4,5,6\}$, perturbed 50 times each by a perturbation factor applied with 4 kinds of probability distributions with 3 sizes of potentially large error, and 3 intervals for small errors (500×4×50×4×3×3 = 3,600,000). The results presented in Table 4 were obtained from 2,700,000 iterations i.e. 250 various PVs of 6 sizes i.e. $n\in \{4,5\dots,9\}$, perturbed 50 times each by perturbation factor applied with 4 kinds of probability distributions with 3 sizes of a possible large error, and 3 intervals for small errors (250×6×50×4×3×3 = 2,700,000).

The results presented in Tables 3 & 4 provide very significant information about possible distortions of PRs obtained via any PDM. They can be concluded in the following way: $AAE$ & $MAD$ are different for various PCM sizes i.e. smaller for larger PCMs and vice versa, larger for smaller PCMs; for smaller PCMs, inverse linear scale gives smaller possible maximal errors in relation to other examined scales; for larger PCMs, Saaty's linear scale outperform other examined scales, providing smaller possible maximal errors; due to possible PVs distortions. The presented errors should be taken into account during a standard prioritization procedure i.e. alternatives ranking as they can indicate a probability of possible ranking alteration. Considering the latter conclusion as crucial for effective MADM via PCs, it must be further developed in detail. Hence, the following examination results will be presented herein for the selected scales and designated sizes of PCMs together with the example development of their importance.

## Discussion of results

A very long discussion concerning the Rank Reversal Phenomenon (RRP) during PCs particularly regarding applications of AHP, is reported in the subject's literature, see e.g. [211,212]. It may be concluded that the most frequent reasons of RRP inclusion, yet are not limited to synthesis procedure, see e.g. [213–215], normalization procedure, see e.g. [216–218], criteria weights, see e.g. [219,220], methods misuse, see e.g. [202], decisional process uncertainty, see

**Step 1**
- Randomly i.e. with the application of the uniform distribution, generate the priority vector $v = [v_1, \ldots, v_n]^T$ of assigned size $[n \times 1]$ and related perfect $MPR(v) = [v_{ij}]_{nxn}$

**Step 2**
- Randomly i.e. with the application of the uniform distribution, select an element $v_{xy}$ for $x<y$ of $MPR(v)$ and replace it with $v_{xy}\varepsilon_s$ where $\varepsilon_s$ represents a possible significant error, and is applied in equal proportions as $\varepsilon_s$=1, $\varepsilon_s$=3, and $\varepsilon_s$=5.

**Step 3**
- For each other element $v_{ij}$, where $i < j \leq n$ select a value $\varepsilon_{ij}$ for the relatively small error and replace the element $v_{ij}$ with the element $v_{ij}\varepsilon_{ij}$ where $\varepsilon_{ij}$ is drawn consecutively with use of *gamma*, *log-normal*, *truncated normal*, and *uniform* distributions from the following intervals applied in equal proportions i.e. $\varepsilon_{ij} \in [0.3, 1.7]$, $\varepsilon_{ij} \in [0.6, 1.4]$, and $\varepsilon_{ij} \in [0.9, 1.1]$.

**Step 4**
- Round all values of $v_{ij}\varepsilon_{ij}$ for $i < j$ to the nearest value of the designated scale in order to obtain $PCM_S(w)$.

**Step 5**
- Replace selected elements of $PCM_S(w)$, i.e. all $w_{ij}$ for $i > j$, with the fraction $1/w_{ij}$ in order to obtain $PCM_{SR}(w)$.

**Step 6**
- Calculate values of the selected Consistency Indices (CI) for $PCM_{SR}(w)$. Remember computed values as one record.

**Step 7**
- Derive the priority vector $w$ for $PCM_{SR}(w)$ with the application of the selected PDM. Then, compute AAE and MAD between $w$ and $v$. Remember computed values as one record.

**Step 8**
- Repeat $S_n$ times *Steps* from 2 to 7.

**Step 9**
- Repeat $S_m$ times *Steps* from 1 to 8.

**Step 10**
- Save *all* records within the one database file.

**Chart 1. Simulation algorithm applied for the research.** Steps relate to descriptions in Chart 1.

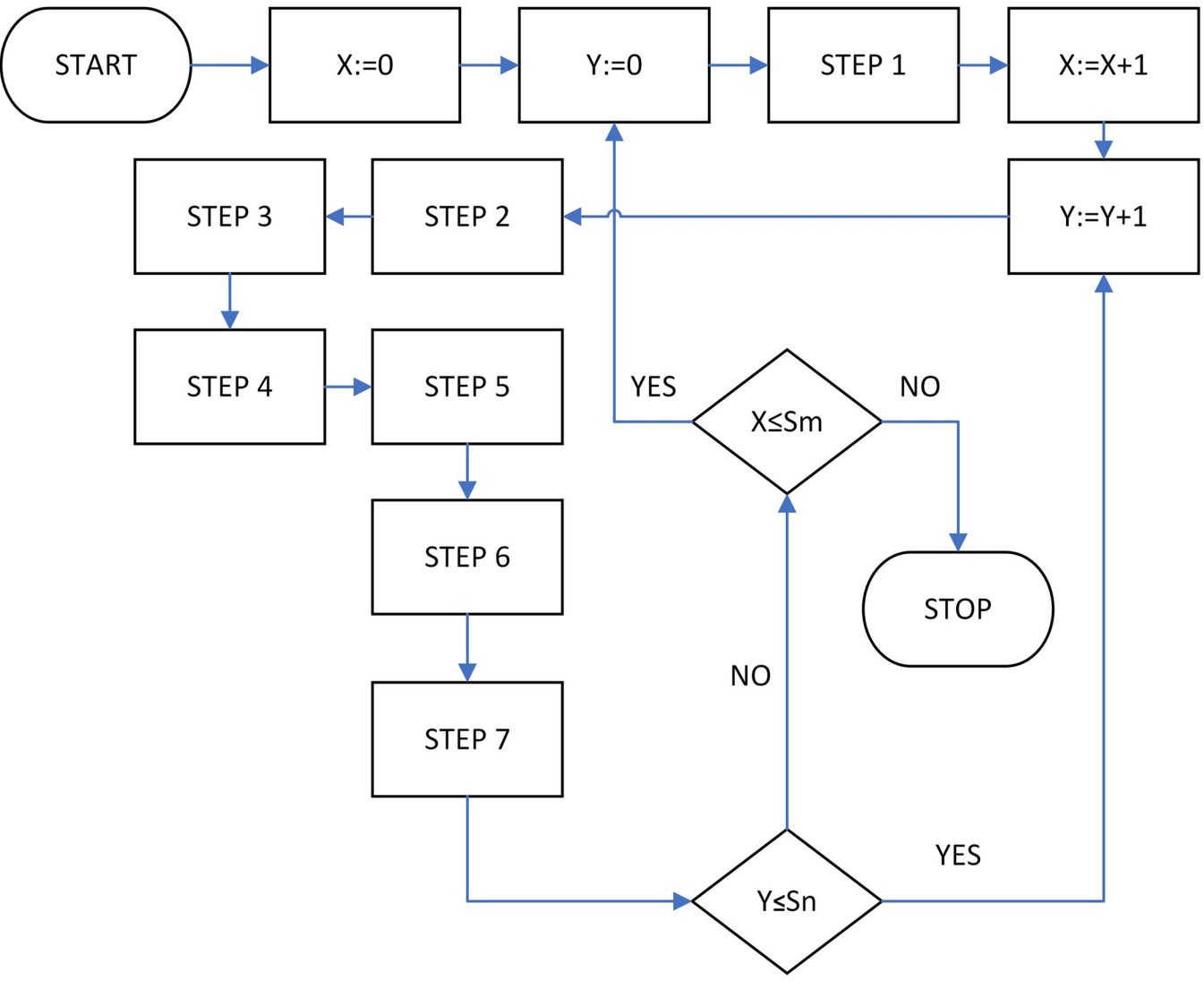

**Fig 1. Flowchart of simulation algorithm applied for the research.**

e.g. [221], and structural dependency among the criteria and alternatives, see e.g. [222,223]. However, it must be emphasized that from the best acquired knowledge from the subject's literature analysis, none so far have published research considering this phenomenon from the perspective of the inexactitude of the prioritization process. Indeed, all choices made so far with the application of the AHP have been made on the basis of the AHP's final ranking which has been taken as established, hence with disregard of possible estimation errors.

In order to grasp a new perspective in this issue, the known example of RRP is proposed to be reexamined [217,224]. In this example, presented for the first time by Belton & Gear [217], the hypothetic problem is analyzed via the AHP which is structured as the three criteria and three alternatives framework. It is assumed that all considered criteria are of equal importance. It is also assumed that DM's judgments concerning the problem's alternative solutions are perfectly consistent and based on the AHP's standard Saaty's linear preference scale. The tables comprising considered PCMs and their related PVs are presented below (Figs 2–5).

**Table 3. Mean values and $p$–Quantiles of _AAE_ for selected scales.**

| PCM size | Scale type | $p$–Quantiles | | | | Mean |
|---|---|---|---|---|---|---|
| | | $p = 0.5$ | $p = 0.9$ | $p = 0.95$ | $p = 0.99$ | |
| $n \in \{3,4,5,6\}$ | Geometric | 0.0157 | 0.0569 | 0.1649 | 0.2434 | 0.0308 |
| | Inverse linear | 0.01485 | 0.10895 | 0.1623 | 0.1978 | 0.0343 |
| | Saaty's linear | 0.0235 | 0.1432 | 0.1861 | 0.3335 | 0.04825 |
| $n \in \{4,5\ldots,9\}$ | Geometric | 0.0072 | 0.0204 | 0.0451 | 0.0542 | 0.01175 |
| | Inverse linear | 0.0162 | 0.0314 | 0.0491 | 0.0563 | 0.0171 |
| | Saaty's linear | 0.0120 | 0.0212 | 0.0212 | 0.0436 | 0.0131 |

After application of the standard AHP's synthesis procedure, the following aggregated PV can be obtained: $p = [0.4512, 0.4697, 0.0791]^T$, which provides the following DM's preference order $p_2 \succ p_1 \succ p_3$.

Next, it is assumed that a new alternative is added to the problem while all previous assumptions concerning the problem structure and DM's judgments remain the same. The accordingly modified tables comprising of extended PCMs and their related PVs are presented below (Figs 6–8).

This time, also after application of the standard AHP's synthesis procedure, the new aggregated PV is obtained: $d = [0.3654, 0.2889, 0.0568, 0.2889]^T$, which provides the following new DM's preference order $d_1 \succ d_2 \equiv d_4 \succ d_3$, which is different than the previous version i.e. $p_2 \succ p_1 \succ p_3$. Despite there being no change in the relative preferences with regard to $A_1$ versus $A_2$, the final preference order between these two elements has been reversed. However, the latter conclusion is based entirely on the intrinsic and purely heuristic assumption about the established character of received results. In order to indicate its erroneous nature, the following table (Table 5) is presented with results of Monte Carlo simulations for particular number of alternatives and Saaty's linear preference scale with the application of the simulation algorithm presented earlier in this research paper (Chart 1). Results presented in Tables 5 and 6 are based on over 10,000 asymptotically consistent PCMs which were obtained out of 900,000 iterations i.e. 500 various PVs, perturbed 50 times each by a perturbation factor applied with 4 kinds of probability distributions with 3 sizes of a possible big error, and 3 intervals for small errors (500×50×4×3×3 = 900,000).

Let the earlier presented PV i.e. $p = [0.4512, 0.4697, 0.0791]^T$ be now reconsidered. As it has been already indicated, assuming its established values as fixed, it provides the following DM's preference order $p_2 \succ p_1 \succ p_3$. However, taking into consideration the fact that the considered PV is only the possible estimate of some true PV which remains unknown, the ranking of its PRs should be preceded by an analysis of possible errors indicated in Tables 5 and 6, as its true PRs' values may fluctuate around considered errors. Hence, for this particular PV, for example,

**Table 4. Mean values and $p$–Quantiles of _MAD_ for selected scales.**

| PCM size | Scale type | $p$–Quantiles | | | | Mean |
|---|---|---|---|---|---|---|
| | | $p = 0.5$ | $p = 0.9$ | $p = 0.95$ | $p = 0.99$ | |
| $n \in \{3,4,5,6\}$ | Geometric | 0.0106 | 0.0368 | 0.1229 | 0.1548 | 0.02097 |
| | Inverse linear | 0.0075 | 0.0678 | 0.1437 | 0.1437 | 0.02215 |
| | Saaty's linear | 0.0198 | 0.0930 | 0.1185 | 0.3269 | 0.0349 |
| $n \in \{4,5\ldots,9\}$ | Geometric | 0.0056 | 0.0251 | 0.0257 | 0.0451 | 0.00999 |
| | Inverse linear | 0.0097 | 0.0305 | 0.0314 | 0.0491 | 0.0133 |
| | Saaty's linear | 0.0109 | 0.0182 | 0.0212 | 0.0212 | 0.0112 |

$$\begin{array}{ccccc} Goal & C_1 & C_2 & C_3 & w_G \\ \begin{matrix} C_1 \\ C_2 \\ C_3 \end{matrix} & \begin{bmatrix} 1 & 1 & 1 \\ 1 & 1 & 1 \\ 1 & 1 & 1 \end{bmatrix} & & \rightleftarrows & \begin{bmatrix} \frac{1}{3} \\ \frac{1}{3} \\ \frac{1}{3} \end{bmatrix} \end{array}$$

**Fig 2. The PCM of criteria with regard to the goal, and its related PV.**

the median of its $MPE_{ME} = 0.01384 + 0.01087 = 0.02471$, the mean of its $MPE_{MN} = 0.05376 + 0.02998 = 0.08374$, and the 0.99 quantile of its $MPE_{Q_{0.99}} = 0.23334 + 0.11667 = 0.35001$. Thus, the true PRs' values of this particular PV may fluctuate plus-minus 0.02471 (if the medians of possible errors are taken into consideration), or 0.08374 (if the means of possible errors are taken into consideration), or even 0.35001 (if the 0.99 quantiles of possible errors are taken into consideration), etc. However, the difference between $p_1$ and $p_2$ merely equals $0.4697 - 0.4512 = 0.0187$ which is even smaller than a double 0.01 quantile of $AAE = 0.01384 \times 2 = 0.02768 > 0.0187$ for this particular PV. It means that there is an average 99% probability that the true preference order which this PV provides may be equally well denoted by the sequence $p_1 \succ p_2 \succ p_3$. Hence, in comparison with the preference order indicated in the example by the second PV i.e. $d_1 \succ d_2 \equiv d_4 \succ d_3$, which by the way also may fluctuate, a rank reversal is not observed.

It is believed that every ranking designated by any PV, obtained or derived via the pairwise comparisons process, should be evaluated from the above presented perspective. It is why, Tables 7–10 present discussed errors also for other selected preference scales and various sizes of consistent PCMs.

When the in/consistency in Pairwise Comparisons (PCs) is taken into consideration as the subarea of the MADM scientific field, it presumably may be perceived as the most exploiting topic in this research area. A variety of models have been proposed to address inconsistency issues, see e.g. [78–85]. Certainly, issues related to PCs in/consistency have many repercussions

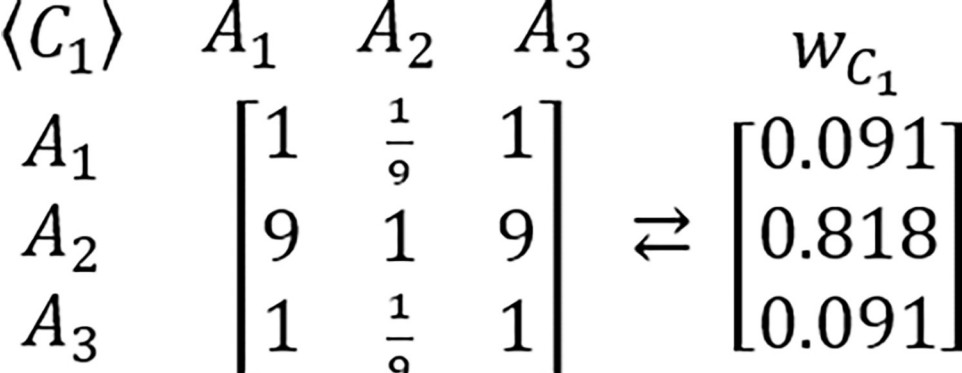

$$\begin{array}{ccccc} \langle C_1 \rangle & A_1 & A_2 & A_3 & w_{C_1} \\ \begin{matrix} A_1 \\ A_2 \\ A_3 \end{matrix} & \begin{bmatrix} 1 & \frac{1}{9} & 1 \\ 9 & 1 & 9 \\ 1 & \frac{1}{9} & 1 \end{bmatrix} & & \rightleftarrows & \begin{bmatrix} 0.091 \\ 0.818 \\ 0.091 \end{bmatrix} \end{array}$$

**Fig 3. The PCM of alternatives with regard to the first criterion, and its related PV.**

$$\langle C_2 \rangle \quad \begin{array}{ccc} A_1 & A_2 & A_3 \end{array} \qquad w_{C_2}$$

$$\begin{array}{c} A_1 \\ A_2 \\ A_3 \end{array} \begin{bmatrix} 1 & 9 & 9 \\ \frac{1}{9} & 1 & 1 \\ \frac{1}{9} & 1 & 1 \end{bmatrix} \quad \rightleftarrows \quad \begin{bmatrix} 0.818 \\ 0.091 \\ 0.091 \end{bmatrix}$$

**Fig 4. The PCM of alternatives with regard to the second criterion, and its related PV.**

$$\langle C_3 \rangle \quad \begin{array}{ccc} A_1 & A_2 & A_3 \end{array} \qquad w_{C_3}$$

$$\begin{array}{c} A_1 \\ A_2 \\ A_3 \end{array} \begin{bmatrix} 1 & \frac{8}{9} & 8 \\ \frac{9}{8} & 1 & 9 \\ \frac{1}{8} & \frac{1}{9} & 1 \end{bmatrix} \quad \rightleftarrows \quad \begin{bmatrix} 0.444 \\ 0.5 \\ 0.056 \end{bmatrix}$$

**Fig 5. The PCM of alternatives with regard to the third criterion, and its related PV.**

$$\langle C_1 \rangle \quad \begin{array}{cccc} A_1 & A_2 & A_3 & A_4 \end{array} \qquad w_{C_1}$$

$$\begin{array}{c} A_1 \\ A_2 \\ A_3 \\ A_4 \end{array} \begin{bmatrix} 1 & \frac{1}{9} & 1 & \frac{1}{9} \\ 9 & 1 & 9 & 1 \\ 1 & \frac{1}{9} & 1 & \frac{1}{9} \\ 9 & 1 & 9 & 1 \end{bmatrix} \quad \rightleftarrows \quad \begin{bmatrix} 0.05 \\ 0.45 \\ 0.05 \\ 0.45 \end{bmatrix}$$

**Fig 6. The extended PCM of alternatives with regard to the first criterion, and its related PV.**

$$\langle C_2 \rangle \quad \begin{array}{cccc} A_1 & A_2 & A_3 & A_4 \end{array} \qquad w_{C_2}$$

$$\begin{array}{c} A_1 \\ A_2 \\ A_3 \\ A_4 \end{array} \begin{bmatrix} 1 & 9 & 9 & 9 \\ \frac{1}{9} & 1 & 1 & 1 \\ \frac{1}{9} & 1 & 1 & 1 \\ \frac{1}{9} & 1 & 1 & 1 \end{bmatrix} \rightleftarrows \begin{bmatrix} 0.75 \\ 0.083 \\ 0.083 \\ 0.083 \end{bmatrix}$$

**Fig 7. The extended PCM of alternatives with regard to the second criterion, and its related PV.**

in various types of modelling scenarios i.e. the inconsistency reduction of reciprocal Pairwise Comparison Matrices (PCMs) with high levels of inconsistency, see e.g. [72,86–91]; deriving appropriate consistency thresholds for non/reciprocal PCMs, see e.g. [10,74,92–96]; completing of incomplete reciprocal PCMs, see e.g. [97–102]; and aggregating of individual reciprocal PCMs in relation to Group Decision Making (GDM) aspects, see e.g. [75,103–110], and PCMs in/consistency relation to credibility of Priority Vectors (PV) derived from PCMs with the application of various Priorities Deriving Methods (PDMs). The examination objective in the latter area of research is the uncertainty related to the inexactitude of prioritization based on derived PVs.

It should be realized here that there are three significantly different notions: the PCM consistency perceived from the perspective of its definition, see hereafter D[3], and expressed by the specific inconsistency index value; the consistency of decision makers, i.e. their trustworthiness, reflected by the number and size of their judgments discrepancies, and; the PCM applicability for estimation of decision makers' priorities in the way that leads to minimization of their estimation errors.

As it seems the third issue is probably the most important problem in the contemporary arena of the MADM theory concerning AHP, and the only way to examine that phenomena is through computer simulations.

$$\langle C_3 \rangle \quad \begin{array}{cccc} A_1 & A_2 & A_3 & A_4 \end{array} \qquad w_{C_3}$$

$$\begin{array}{c} A_1 \\ A_2 \\ A_3 \\ A_4 \end{array} \begin{bmatrix} 1 & \frac{8}{9} & 8 & \frac{8}{9} \\ \frac{9}{8} & 1 & 9 & 1 \\ \frac{1}{8} & \frac{1}{9} & 1 & \frac{1}{9} \\ \frac{9}{8} & 1 & 9 & 1 \end{bmatrix} \rightleftarrows \begin{bmatrix} 0.296 \\ 0.333 \\ 0.037 \\ 0.333 \end{bmatrix}$$

**Fig 8. The extended PCM of alternatives with regard to the third criterion, and its related PV.**

**Table 5. Mean values and $p$–Quantiles of $AAE$ for Saaty's linear scale.**

| PCM size | $p$–Quantiles | | | Mean |
|---|---|---|---|---|
| | $0.01{\leq}p{\leq}0.1$ | $p = 0.5$ | $0.9{\leq}p{\leq}0.99$ | |
| 3 | 0.01384 | 0.01384 | 0.23334 | 0.05376 |
| 4 | 0.00617 | 0.01177 | 0.02985 | 0.01279 |
| 5 | 0.00664 | 0.01170 | 0.03825 | 0.01473 |
| 6 | 0.00250 | 0.01100 | 0.02084 | 0.01085 |
| 7 | 0.00762 | 0.01214 | 0.01823 | 0.01247 |
| 8 | 0.00614 | 0.00763 | 0.01277 | 0.00830 |
| 9 | 0.00601 | 0.00941 | 0.01209 | 0.00949 |

The PCMs in/consistency relation to credibility of Priority Vectors (PV) derived from PCMs with the application of various Priorities Deriving Methods (PDMs) constitutes the key issue in a few research studies e.g. [51,76,77,111–117]. As it was previously stated the examination objective in this area of research is the uncertainty related to the inexactitude of prioritization based on derived PVs. However, only few research studies examine this problem from the perspective of PCM applicability for credible designation of decision maker's (DM) priorities in the way that leads to minimization of the prioritization uncertainty related to possible, and sometimes very probable, ranking fluctuations. This problem constitutes the primary area of interest for this research paper as no other research study was thus far identified that examines this problem from the perspective of consistent PCMs.

So far, this concept has been studied only from the perspective of inconsistent PCMs, see e.g. [51,76,77,111,112,114,117]. Hence, a research gap was identified. Thus, the objective of this research paper was to fill in this scientific gap. The research findings have serious repercussions in relation to prioritization quality with the application of PCs methodology, mostly in relation to the interpretation and reliability evaluation of prioritization results. Firstly, the research study outcome changes the perspective of the rank reversal phenomenon, which shed new light on many research studies that have been presented in the subject's literature for many decades. Secondly, the research study results throw new light on the discussion concerning the fuzziness of AHP's results. Last but not least, the effect of the research opens the unique opportunity to evaluate the prioritization outcome obtained within the process of consistent PCs from the well-known perspective of statistical hypothesis testing i.e. the probability designation of the chance that accepted ranking results which were considered as correct due to low probability of change may be incorrect, hence they should be rejected, and the probability designation of the chance that rejected ranking results which were considered as incorrect due to high probability of change may be correct and should be accepted. The paramount finding of

**Table 6. Mean values and $p$–Quantiles of $MAD$ for Saaty's linear scale.**

| PCM size | $p$–Quantiles | | | Mean |
|---|---|---|---|---|
| | $0.01{\leq}p{\leq}0.1$ | $p = 0.5$ | $0.9{\leq}p{\leq}0.99$ | |
| 3 | 0.01087 | 0.01087 | 0.11667 | 0.02998 |
| 4 | 0.00449 | 0.01177 | 0.02986 | 0.01194 |
| 5 | 0.00640 | 0.01168 | 0.02633 | 0.01446 |
| 6 | 0.00190 | 0.01370 | 0.02713 | 0.01366 |
| 7 | 0.00941 | 0.01331 | 0.01955 | 0.01414 |
| 8 | 0.00927 | 0.01001 | 0.03017 | 0.01345 |
| 9 | 0.00864 | 0.01269 | 0.03058 | 0.01460 |

**Table 7. Mean values and p–Quantiles of *AAE* for ILS.**

| PCM size | p–Quantiles | | | Mean |
|---|---|---|---|---|
| | $0.01 \leq p \leq 0.1$ | $p = 0.5$ | $0.9 \leq p \leq 0.99$ | |
| 3 | 0.01115 | 0.03035 | 0.17800 | 0.06986 |
| 4 | 0.00316 | 0.00790 | 0.03479 | 0.01224 |
| 5 | 0.00308 | 0.00533 | 0.00908 | 0.00588 |
| 6 | 0.00166 | 0.00652 | 0.00907 | 0.00554 |
| 7 | 0.00213 | 0.00504 | 0.00746 | 0.00473 |
| 8 | 0.00171 | 0.00311 | 0.01099 | 0.00394 |
| 9 | 0.00216 | 0.00333 | 0.00496 | 0.00346 |

**Table 8. Mean values and p–Quantiles of *MAD* for ILS.**

| PCM size | p–Quantiles | | | Mean |
|---|---|---|---|---|
| | $0.01 \leq p \leq 0.1$ | $p = 0.5$ | $0.9 \leq p \leq 0.99$ | |
| 3 | 0.00557 | 0.02547 | 0.09472 | 0.03822 |
| 4 | 0.00179 | 0.00497 | 0.02398 | 0.00733 |
| 5 | 0.00328 | 0.00496 | 0.01340 | 0.00614 |
| 6 | 0.00148 | 0.00777 | 0.01303 | 0.00777 |
| 7 | 0.00284 | 0.00564 | 0.01864 | 0.00789 |
| 8 | 0.00345 | 0.00414 | 0.02344 | 0.00677 |
| 9 | 0.00324 | 0.00530 | 0.00951 | 0.00592 |

**Table 9. Mean values and p–Quantiles of *AAE* for GS.**

| PCM size | p–Quantiles | | | Mean |
|---|---|---|---|---|
| | $0.01 \leq p \leq 0.1$ | $p = 0.5$ | $0.9 \leq p \leq 0.99$ | |
| 3 | 0.01898 | 0.05494 | 0.09091 | 0.05494 |
| 4 | 0.00505 | 0.02393 | 0.09906 | 0.03246 |
| 5 | 0.00663 | 0.00869 | 0.03516 | 0.01273 |
| 6 | 0.00300 | 0.00603 | 0.00982 | 0.00625 |
| 7 | 0.00251 | 0.00426 | 0.00778 | 0.00478 |
| 8 | 0.00321 | 0.00410 | 0.00646 | 0.00418 |
| 9 | 0.00300 | 0.00453 | 0.00525 | 0.00419 |

**Table 10. Mean values and p–Quantiles of *MAD* for GS.**

| PCM size | p–Quantiles | | | Mean |
|---|---|---|---|---|
| | $0.01 \leq p \leq 0.1$ | $p = 0.5$ | $0.9 \leq p \leq 0.99$ | |
| 3 | 0.01471 | 0.04244 | 0.07017 | 0.04244 |
| 4 | 0.00505 | 0.02393 | 0.09762 | 0.03063 |
| 5 | 0.00640 | 0.01034 | 0.04021 | 0.01462 |
| 6 | 0.00387 | 0.00923 | 0.01543 | 0.00913 |
| 7 | 0.00247 | 0.00864 | 0.01065 | 0.00772 |
| 8 | 0.00344 | 0.00459 | 0.01383 | 0.00575 |
| 9 | 0.00340 | 0.00559 | 0.01019 | 0.00646 |

the research is the fact that consistent PCMs provide PVs, which elements cannot be considered as established, but only approximated within certain confidence intervals estimated with a certain level of probability. As problems related to heuristics can be analyzed only via a computer simulation process, because they cannot be mathematically determined, the problem examined in this research paper was examined via Monte Carlo simulations, appropriately coded and executed with the application of Wolfram's Mathematica Software. It is believed that this research findings should be very important and useful for all decision makers and researchers during their problems' examinations that relate to prioritization processes with the application of PCs methodology.

## Final remarks

The conducted research study is a mile stone on the way of interpretation and reliability evaluation of results of the prioritization process with the application of the pairwise comparisons technique. Firstly, the research study results change the perspective of the rank reversal phenomenon, which gives a new light on many research studies that have been presented in the subject's literature for many decades, see e.g. [211–216,218,220,222,223,225,226].

Secondly, the research study results throw a new light on the discussion concerning the fuzziness of the AHP's results. As is known, the creator of the AHP was strictly against further fuzzifying of the AHP's results, see e.g. [141,142]. Yet, there are many fuzzy AHP applications in the literature, and they are very popular, see e.g. [138,227–229].

Last but not least, the effect of this research opens a unique opportunity to evaluate the prioritization outcome obtained within the process of consistent pairwise comparisons from the well-known perspective from the statistical hypothesis testing theory i.e. the probability of making the error of the first and the second kind i.e. accepting the ranking results which are not valid as they are prone to change, and rejecting the ranking results which may be valid. This has significant repercussions for many application-oriented research papers which apply the pairwise comparisons method, see e.g. [139,140,230–233].

No research study is perfect when it comes to the reality of the physical world. Hence, this research study has also its limitations. They concern mostly the way of the human judgments imitation process i.e. simulation of human judgment's errors. The nature of human judgments can only be represented as a realization of some random process in accordance with the assumed probability distribution of the perturbation factor e.g. uniform, gamma, truncated normal, log-normal, etc., see e.g. [115,234]. This is the main limitation of this research because this process is only a stochastic process generated by computer algorithms.

Future direction of the research may challenge these limitations as well include other preference scales which were not examined in this research paper.

## Acknowledgments

The author would like to acknowledge the support of two independent Referees for their benevolence and a very smooth reviewing process, especially Dr. Bagh Ali who decided to reveal the identity, for very constructive comments which improved the first version of the manuscript. The author also appreciates valuable comments made by the Editor which improved both the second, and the third version of the manuscript.

## Author Contributions

**Conceptualization:** Pawel Tadeusz Kazibudzki.

**Data curation:** Pawel Tadeusz Kazibudzki.

**Formal analysis:** Pawel Tadeusz Kazibudzki.

**Funding acquisition:** Pawel Tadeusz Kazibudzki.

**Investigation:** Pawel Tadeusz Kazibudzki.

**Methodology:** Pawel Tadeusz Kazibudzki.

**Project administration:** Pawel Tadeusz Kazibudzki.

**Resources:** Pawel Tadeusz Kazibudzki.

**Validation:** Pawel Tadeusz Kazibudzki.

**Writing – original draft:** Pawel Tadeusz Kazibudzki.

**Writing – review & editing:** Pawel Tadeusz Kazibudzki.

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
