## [Decision Letter · Decision Letter 0]

14 Jun 2023

PONE-D-23-12219On Uncertainty Related to the Inexactitude of Prioritization based on Consistent Pairwise ComparisonsPLOS ONE

Dear Dr. Kazibudzki,

Thank you for submitting your manuscript to PLOS ONE. After careful consideration, we feel that it has merit but does not fully meet PLOS ONE’s publication criteria as it currently stands. Therefore, we invite you to submit a revised version of the manuscript that addresses the points raised during the review process.

We look forward to receiving your revised manuscript.

Kind regards,

Muhammad Hashim, PhD

Academic Editor

PLOS ONE

Journal Requirements:

ON THE SIMILARITY AMONG PRIORITY DERIVING METHODS FOR THE AHP - http://mail.isahp.org/uploads/027_001.pdf

In your revision ensure you cite all your sources (including your own works), and quote or rephrase any duplicated text outside the methods section. Further consideration is dependent on these concerns being addressed.

3. Please note that PLOS ONE has specific guidelines on code sharing for submissions in which author-generated code underpins the findings in the manuscript. In these cases, all author-generated code must be made available without restrictions upon publication of the work. 

Please review our guidelines at https://journals.plos.org/plosone/s/materials-and-software-sharing#loc-sharing-code and ensure that your code is shared in a way that follows best practice and facilitates reproducibility and reuse.

"This work was ﬁnancially supported by the Opole University of Technology under GAMMA project no. 152/22. The APC was partially funded by Opole University of Technology, Poland.

The APC funders had no role in the design of the study; in the collection, analyses, or interpretation of data; in the writing of the manuscript, or in the decision to publish the results."

We note that one or more of the authors is affiliated with the funding organization, indicating the funder may have had some role in the design, data collection, analysis or preparation of your manuscript for publication; in other words, the funder played an indirect role through the participation of the co-authors. If the funding organization did not play a role in the study design, data collection and analysis, decision to publish, or preparation of the manuscript and only provided financial support in the form of authors' salaries and/or research materials, please do the following:

(1) Review your statements relating to the author contributions, and ensure you have specifically and accurately indicated the role(s) that these authors had in your study. These amendments should be made in the online form.

(2) Confirm in your cover letter that you agree with the following statement, and we will change the online submission form on your behalf: 

**Additional Editor Comments:**

Major improvements are required to bring your paper to the next level. I believe that the paper’s findings will be beneficial to related stakeholders in the field.

Reviewers' comments:

Reviewer's Responses to Questions

**Comments to the Author**

1. Is the manuscript technically sound, and do the data support the conclusions?

Reviewer #1: Yes

Reviewer #2: Yes

2. Has the statistical analysis been performed appropriately and rigorously? 

Reviewer #1: N/A

Reviewer #2: Yes

3. Have the authors made all data underlying the findings in their manuscript fully available?

Reviewer #1: Yes

Reviewer #2: Yes

4. Is the manuscript presented in an intelligible fashion and written in standard English?

Reviewer #1: Yes

Reviewer #2: Yes

5. Review Comments to the Author

Reviewer #1: . I have some minor concern:

1. Add some main finding in the abstract and also add one or two lines about singnificance of present study which show why you choose this topic.

2. The title is not uniform because some words are start capital and some capital and small letter both.

3. The introduction. The authors should make it clear why this study is novel.

4. In the Introduction, the literature review was not logically organized and all literatures cited seem separate descriptions without connections. The readers can't know what the state-of-art methodologies or gaps the current study plans to resolve or fill, and how significant or what contribution the current study is?. Rewrite the last paragraph of introduction section and make it concise which contains the objective, novelty, motivation, and why you choose this problem..

5. Must add future direction of your research..

6. . Consequently, it is important to call your attention to the fact that all the sections must be logically connected. Based on the fact that the outcome of a research, after published, is cited and built-upon in the world, it is very important to improve the quality of presentation. Consequently, you need to

(a) connect the title to the abstract

(b) connect the abstract to the introduction

(c) prepare the introduction to accurately lead to the analysis of results.

(d) divide the discussion of results into (i) analysis of results and (ii) discussion of results.

(e) conclude the report based on the facts you have analyzed and discussed.

Comment: Do the needful. Follow the pattern above to connect all the sections.

Also the authors are advised to update the manuscript before final submission as it contains some typos and grammatical errors.

Reviewer #2: The manuscript appears to be well written in standard English. Statistical analyses were also performed rigorously. However, all data underlying the findings were not found in the manuscript. Recent literature were reviewed in the manuscript but a section titled "Systematic Review of Literature" seems to be missing.

6. PLOS authors have the option to publish the peer review history of their article (what does this mean?). If published, this will include your full peer review and any attached files.

Reviewer #1: No

Reviewer #2: No

---

## [Author Response · Author response to Decision Letter 0]

23 Jun 2023

Please find my responses to reviewers and editor related comments in the attached files i.e. my Cover Letter and Response to Reviewers file.

---

## [Decision Letter · Decision Letter 1]

24 Jul 2023

PONE-D-23-12219R1On Uncertainty Related to the Inexactitude of Prioritization Based on Consistent Pairwise ComparisonsPLOS ONE

Dear Dr. Kazibudzki,

Thank you for submitting your manuscript to PLOS ONE. After careful consideration, we feel that it has merit but does not fully meet PLOS ONE’s publication criteria as it currently stands. Therefore, we invite you to submit a revised version of the manuscript that addresses the points raised during the review process.

We look forward to receiving your revised manuscript.

Kind regards,

Muhammad Hashim, PhD

Academic Editor

PLOS ONE

Journal Requirements:

Additional Editor Comments:

Following points need to be improved;

1. The title of study should be like that ‘The uncertainty in decision making process related to the inexactitude of prioritization based on consistent pairwise comparisons’

2. The abstract part should be revised by adding concrete objective, problem statement and originality of proposed study. Abstract part is very confusing and does not make any sense what authors have done and contributed. At the end of abstract please describe who will be the target beneficiaries of your study?

3. In the introduction part second line what do you mean by ‘a human mind can be lost when unsupported during a decision making process’? a human mind can be lost….correct it. Same type of mistakes found in the rest of manuscript. Introduction part should describe the brief overview of your topic along significance and approaches/algorithm authors developed. The paragraphs are not consistent.

4. Authors used Monte Carlo Simulation algorithm. What is new in your study? Monte Carlo Simulation is an old fashioned approach/tool and a lot of researchers have applied.

5. Review of literature is not relevant to the study. Authors must add relevant studies of fuzzy AHP and traditional AHP method as well as Monte Carlo Simulation algorithm.

6. Formal notations should be written in equation form containing proper numbers and descriptions.

7. Authors added Examination Method in their study but did not describe who develop this method? Why did you use this method? Add some properties and significance of this method?

8. The problem examinations scenario is poorly written and not highlighting the genuine issue addressed in this research.

9. Figure 1. Simulation algorithm should be in Flowchart form.

10. Discussion of results is not compared with existing literature/ studies. How better are your results as compared to other studies?

11. The implications of research should be added before conclusion part based on your results.

12. The conclusion part is not in literary form. It must be improved.

13. Finally, proofreading from professional are required for the entire manuscript.

Reviewers' comments:

Reviewer's Responses to Questions

**Comments to the Author**

1. If the authors have adequately addressed your comments raised in a previous round of review and you feel that this manuscript is now acceptable for publication, you may indicate that here to bypass the “Comments to the Author” section, enter your conflict of interest statement in the “Confidential to Editor” section, and submit your "Accept" recommendation.

Reviewer #1: All comments have been addressed

Reviewer #2: All comments have been addressed

2. Is the manuscript technically sound, and do the data support the conclusions?

Reviewer #1: Yes

Reviewer #2: Yes

3. Has the statistical analysis been performed appropriately and rigorously? 

Reviewer #1: N/A

Reviewer #2: Yes

4. Have the authors made all data underlying the findings in their manuscript fully available?

Reviewer #1: Yes

Reviewer #2: Yes

5. Is the manuscript presented in an intelligible fashion and written in standard English?

Reviewer #1: Yes

Reviewer #2: Yes

6. Review Comments to the Author

Reviewer #1: The authors revised the manuscript according to the wise suggestion so the revised manuscript is suitable for publication.

Reviewer #2: (No Response)

7. PLOS authors have the option to publish the peer review history of their article (what does this mean?). If published, this will include your full peer review and any attached files.

Reviewer #1: **Yes: **Dr.Bagh Ali

Reviewer #2: No

---

## [Author Response · Author response to Decision Letter 1]

3 Aug 2023

Responses to Reviewers and the Academic Editor

The content of this document will be devoted exclusively to suggestions made by the editor as the manuscript at its current stage is fully and unconditionally accepted by two independent reviewers. They both claim that the manuscript is suitable for publication as it stands –

Reviewer #1: All comments have been addressed; The authors revised the manuscript according to the wise suggestion so the revised manuscript is suitable for publication.

Reviewer #2: All comments have been addressed. 

The author is confused that the editor reveals content related suggestions concerning the manuscript at the current stage of the reviewing process and not earlier. The author understands that it happens for a reason, hence he will try to use this opportunity to further improve the content of the research paper. However, the author does not agree with all suggestions made by the editor as he believes that some of them, if applied, could actually either confuse a prospective reader about the content of the manuscript or even diminish the value of the research paper as it currently stands. 

Additional Editor Comments:

Following points need to be improved;

1. The title of study should be like that ‘The uncertainty in decision making process related to the inexactitude of prioritization based on consistent pairwise comparisons’;

The author does not agree with the editor about this part of his suggestions. 

Firstly, it is not the time, at the current stage of the reviewing process, to change the title of the manuscript after its full and unconditional acceptance by two independent reviewers. Thus, the author is seriously anxious that the prospective title change could lead to another round of redundant reviews devoted to the ‘new’ perspective of the paper.

Secondly, the suggestion of the editor is incorrect from the perspective of the article’s content. Uncertainty refers to epistemic situations involving imperfect or unknown information. It applies to predictions of future events, to physical measurements that are already made, or to the unknown. Thus, it is a state of limited knowledge where it is impossible to exactly describe the existing situation, a future outcome, or more than one possible output. The title of the article clearly suggests that it refers to the uncertainty related to the inexactitude of prioritization based on consistent pairwise comparisons, and this is the point. The editor’s suggestion refers to the uncertainty of the decision making process which takes into account the inexactitude of prioritization based on consistent pairwise comparisons. In the opinion of the author the proposition does not reflect the same perspective, as a decision making process is the more complex activity which may involve and/or be associated with prioritization but it does not refer explicitly to the issue of prioritization inexactitude i.e. ranking perturbations. Hence, the title change is not possible in this moment.

2. The abstract part should be revised by adding concrete objective, problem statement and originality of proposed study. Abstract part is very confusing and does not make any sense what authors have done and contributed. At the end of abstract please describe who will be the target beneficiaries of your study?

The author decided to use this opportunity and entirely rewrite the abstract although it was already accepted by two independent reviewers. The author fully met all suggestions of the editor. Quoting the editor’s former comment made during the first revision it is believed that “the paper’s findings will be beneficial to related stakeholders in the field”. 

3. In the introduction part second line what do you mean by ‘a human mind can be lost when unsupported during a decision making process’? a human mind can be lost….correct it. Same type of mistakes found in the rest of manuscript. Introduction part should describe the brief overview of your topic along significance and approaches/algorithm authors developed. The paragraphs are not consistent.

It is just the expression that a native speaker introduced to simplify the manuscript content. The author decided to rephrase the questionable part of the sentence and present it in the following form: “humans are not capable of dealing accurately with more than about seven (±2) things at a time (the human brain is limited in its short term memory capacity, its discrimination ability and its range of perception)”. The author does not feel that this kind of phrases constitute some kind of mistakes in the manuscript. They are just phrases commonly used and accepted in American popular scientific language. The introduction part describes the brief overview of the manuscript’s topic along with its significance. The author decided to improve its content and introduce some new paragraphs i.e.

“The PCMs in/consistency relation to credibility of Priority Vectors (PV) derived from PCMs with the application of various Priorities Deriving Methods (PDMs) constitutes the key issue in a few research studies e.g. [1–10]. The examination objective in the latter area of research is the uncertainty related to the inexactitude of prioritization based on derived PVs. However, only few research studies examine this problem from the perspective of PCM applicability for credible designation of decision maker’s (DM) priorities in the way that leads to minimization of the prioritization uncertainty related to possible, and sometimes very probable, ranking fluctuations. This problem constitutes the primary area of interest for this research paper as no research study was thus far identified that examines this problem from the perspective of consistent PCMs.”;

“Hence, a research gap was identified. Thus, the objective of this research paper is to fill in this scientific gap. The research findings have serious repercussions in relation to prioritization quality with the application of PCs methodology, mostly in relation to the interpretation and reliability evaluation of prioritization results. Firstly, the research study outcome changes the perspective of the rank reversal phenomenon, which shed new light on many research studies that have been presented in the subject’s literature for many decades. Secondly, the research study results throw new light on the discussion concerning the fuzziness of AHP’s results. Last but not least, the effect of the research opens the unique opportunity to evaluate the prioritization outcome obtained within the process of consistent PCs from the well-known perspective of statistical hypothesis testing i.e. the probability designation of the chance that accepted ranking results which were considered as correct due to low probability of change may be incorrect, hence they should be rejected, and the probability designation of the chance that rejected ranking results which were considered as incorrect due to high probability of change may be correct and should be accepted. The paramount finding of the research is the fact that consistent PCMs provide PVs, which elements cannot be considered as established, but only approximated within certain confidence intervals estimated with a certain level of probability. As problems related to heuristics can be analyzed only via a computer simulation process, because they cannot be mathematically determined, the problem examined in this research paper is examined via Monte Carlo simulations, appropriately coded and executed with the application of Wolfram’s Mathematica Software. It is believed that this research findings should be very important and useful for all decision makers and researchers during their problems‘ examinations that relate to prioritization processes with the application of PCs methodology.”

The paragraphs were evaluated by the professional proof reader and found to be consistent and appropriately written. This was also concurred by the two independent reviewers after the first revision of this manuscript. 

4. Authors used Monte Carlo Simulation algorithm. What is new in your study? Monte Carlo Simulation is an old fashioned approach/tool and a lot of researchers have applied.

The author does not agree with the editor in this part of his opinion, especially in the part that “Monte Carlo Simulation is an old fashioned approach/tool”. Monte Carlo Simulations are very credible and resourceful method providing very solid information in situations where stochastic problems are considered. Research papers based on Monte Carlo Simulations constitute valuable contribution to the present state of knowledge and they are published nowadays in various prestigious journals e.g. [1,2,10–17]. Hence, it is true that a lot of researchers have been applying this methodology, it is so because Monte Carlo Simulations are very valid and credible source of information. The author already explained during the reviewing process as well in the manuscript in details what is new in its content and why the presented research findings are so important for prospective readers and will be beneficial to related stakeholders in the field. These explanations can be found in the article’s Abstract, Introduction section, and Discussion section. It seems that the editor also noticed the manuscript’s value from that perspective during the first round of the revision process, when he claimed “I believe that the paper’s findings will be beneficial to related stakeholders in the field.” Furthermore, the author is very much confused and anxious as the editor’s questions of the type “What is new in your study?” or statements like “Monte Carlo Simulation is an old fashioned approach/tool” clearly indicate that the manuscript's content begins to be evaluated also on the basis of its novelty and this is the approach which is rather contrary to the PLoS ONE journal’s policy, see e.g. Section - Editorial and Peer Review Process - “The editors make decisions on submissions based on scientific rigor, regardless of novelty”

(source: https://journals.plos.org/plosone/s/editorial-and-peer-review-process). The author would like to believe that both sides are obliged to follow the journal’s policies and requirements as the selection of the journal by the author for prospective publication of his research took place on the bases of explicitly stated policies and requirements of the PLoS ONE journal.

5. Review of literature is not relevant to the study. Authors must add relevant studies of fuzzy AHP and traditional AHP method as well as Monte Carlo Simulation algorithm.

The research paper at the present moment contains 232 references which are explicitly and directly related to the field of study. The relevant references of fuzzy AHP and traditional AHP as well as Monte Carlo Simulations are and have been already quoted in the manuscript. The author does not understands this suggestion as it is unrelated with the current stage and content of the manuscript. If the editor refers to the section “Systematic review of literature” – these few pages contain very important information about various Multi-Attributes Decision Making methods (MADM), and locate the AHP and its developments among other MADM techniques which not necessarily are based on pairwise comparisons what is the key issue in the perspective of this research paper. Hence, the author claims that the literature review is very relevant to the study. This claim is also supported and confirmed by two independent peer reviewers who unconditionally accepted the manuscript in its current stage. The author attracts the attention of the editor that many papers of the editor’s authorship were already included to the manuscript during the first revision, hence their important contribution was not overlooked, see e.g. [18–21]. However, taking into account the editor’s comment this section was improved and expanded too.

6. Formal notations should be written in equation form containing proper numbers and descriptions.

If the editor refers to the section “Formal notations”, it was renamed to “Preliminary remarks”. However, where it was necessary the author also corrected formal notations in the manuscript in order to provide their appropriate form. 

7. Authors added Examination Method in their study but did not describe who develop this method? Why did you use this method? Add some properties and significance of this method?

The author upgraded the manuscript from the suggested viewpoint. All the questions of the editor are now clarified in the manuscript.

8. The problem examinations scenario is poorly written and not highlighting the genuine issue addressed in this research.

This section is now renamed into ‘Description of the simulation concept’ that better reflects its purpose. The section was significantly improved and supplemented with the flowchart reflecting the simulation scenario which was requested by the editor. This section is mainly devoted to description of the Monte Carlo simulation scenario applied in this research paper. However, it should be emphasized herein that the novelty of the entire research does not lay in the data generation algorithm but in the perspective and the way these data are examined and evaluated!! The various versions of the similar data generation algorithms have been applied for many years to similar research, see e.g. [1,2,10–17] and/or e.g. [4,7,8,22,23]. All those examinations were considered to be very credible, solid and valid. 

9. Figure 1. Simulation algorithm should be in Flowchart form.

The author does not think that this suggestion is the must have condition to be applied as many papers present similar algorithms in the form of simple lists and it is still okay as they are successfully published, see e.g. [1,2,4,7,8,22,23]. However, the author decided to use the editor suggestion and present the simulation scenario also in the form of the flowchart. Taking into consideration the fact that neither of the reviewers suggested any corrections in this matter, the flowchart supplements the remained presentation in Fig 1.

10. Discussion of results is not compared with existing literature/ studies. How better are your results as compared to other studies?

This section was improved. However, the results presented in this article are unique in their aspect, what was many times emphasized in the manuscript.

11. The implications of research should be added before conclusion part based on your results.

It was done in the manuscript.

12. The conclusion part is not in literary form. It must be improved.

The Conclusion section was renamed into Final remarks section and significantly improved, including its literary form that was previously disturbed. Thank you for the remark.

13. Finally, proofreading from professional are required for the entire manuscript.

The manuscript was proofread twice by a professional native speaker.

The author would like to believe that changes made to the manuscript will satisfy the editor’s expectations and the manuscript will be approved for publication in the PLoS ONE journal as it was already unconditionally approved by two independent reviewers. Thank you –

Respectfully yours, 

Pawel Tadeusz Kazibudzki, PhD

References

1. Kazibudzki, P.T. An Examination of Performance Relations among Selected Consistency Measures for Simulated Pairwise Judgments. Ann. Oper. Res. 2016, 244, 525–544, doi:10.1007/s10479-016-2131-6.

2. Kazibudzki, P.T. An Examination of Ranking Quality for Simulated Pairwise Judgments in Relation to Performance of the Selected Consistency Measure. Adv. Oper. Res. 2019, 2019, e3574263, doi:10.1155/2019/3574263.

3. Kazibudzki, P.T. On Estimation of Priority Vectors Derived from Inconsistent Pairwise Comparison Matrices. J. Appl. Math. Comput. Mech. 2022, 21, 52–59, doi:10.17512/jamcm.2022.4.05.

4. Kazibudzki, P.T.; Křupka, J. Pairwise Judgments Consistency Impact on Quality of Multi-Criteria Group Decision-Making with AHP. EM Ekon. Manag. 2019, 22, 195–212, doi:10.15240/tul/001/2019-4-013.

5. Kazibudzki, P.T. Redefinition of Triad’s Inconsistency and Its Impact on the Consistency Measurement of Pairwise Comparison Matrix. J. Appl. Math. Comput. Mech. 2016, 15, 71–78, doi:10.17512/jamcm.2016.1.07.

6. Kazibudzki, P.T. The AHP Phenomenon of Rank Reversal Demystified.; December 2022.

7. Grzybowski, A.Z.; Starczewski, T. New Look at the Inconsistency Analysis in the Pairwise-Comparisons-Based Prioritization Problems. Expert Syst. Appl. 2020, 113549, doi:10.1016/j.eswa.2020.113549.

8. Grzybowski, A.Z. New Results on Inconsistency Indices and Their Relationship with the Quality of Priority Vector Estimation. Expert Syst. Appl. 2016, 43, 197–212, doi:10.1016/j.eswa.2015.08.049.

9. Grzybowski, A.Z.; Starczewski, T. Remarks about Inconsistency Analysis in the Pairwise Comparison Technique. In Proceedings of the 2017 IEEE 14th International Scientific Conference on Informatics; November 2017; pp. 227–231.

10. Grzybowski, A.Z.; Starczewski, T. Simulation Analysis of Prioritization Errors in the AHP and Their Relationship with an Adopted Judgement Scale. In Proceedings of the Proceedings; San Francisco, USA, October 23 2018; p. 5.

11. Carmone, F.J.; Kara, A.; Zanakis, S.H. A Monte Carlo Investigation of Incomplete Pairwise Comparison Matrices in AHP. Eur. J. Oper. Res. 1997, 102, 538–553, doi:10.1016/S0377-2217(96)00250-0.

12. Herman, M.W.; Koczkodaj, W.W. A Monte Carlo Study of Pairwise Comparison. Inf. Process. Lett. 57, 25–29.

13. Zahedi, F. A Simulation Study of Estimation Methods in the Analytic Hierarchy Process. Socioecon. Plann. Sci. 1986, 20, 347–354, doi:10.1016/0038-0121(86)90046-7.

14. Wu, H.; Leung, S.-O. Can Likert Scales Be Treated as Interval Scales?—A Simulation Study. J. Soc. Serv. Res. 2017, 43, 527–532, doi:10.1080/01488376.2017.1329775.

15. Zanakis, S.H.; Solomon, A.; Wishart, N.; Dublish, S. Multi-Attribute Decision Making: A Simulation Comparison of Select Methods. Eur. J. Oper. Res. 1998, 107, 507–529, doi:10.1016/S0377-2217(97)00147-1.

16. Winsberg, E. Science in the Age of Computer Simulation; University of Chicago Press: Chicago, IL, 2010; ISBN 978-0-226-90204-3.

17. Kazibudzki, P.T. The Quality of Ranking during Simulated Pairwise Judgments for Examined Approximation Procedures. Model. Simul. Eng. 2019, 2019, e1683143, doi:10.1155/2019/1683143.

18. Nazam, M.; Hashim, M.; Ahmad Baig, S.; Abrar, M.; Ur Rehman, H.; Nazim, M.; Raza, A. Categorizing the Barriers in Adopting Sustainable Supply Chain Initiatives: A Way-Forward towards Business Excellence. Cogent Bus. Manag. 2020, 7, 1825042, doi:10.1080/23311975.2020.1825042.

19. Nazam, M.; Ahmad, J.; Javed, M.K.; Hashim, M.; Yao, L. Risk-Oriented Assessment Model for Project Bidding Selection in Construction Industry of Pakistan Based on Fuzzy AHP and TOPSIS Methods. In Proceedings of the Proceedings of the Eighth International Conference on Management Science and Engineering Management; Xu, J., Cruz-Machado, V.A., Lev, B., Nickel, S., Eds.; Springer: Berlin, Heidelberg, 2014; pp. 1165–1177.

20. Nazam, M.; Hashim, M.; Ahmad, J.; Ahmad, W.; Tahir, M. Selection of Reverse Logistics Operating Channels Through Integration of Fuzzy AHP and Fuzzy TOPSIS: A Pakistani Case. In Proceedings of the Proceedings of the Tenth International Conference on Management Science and Engineering Management; Xu, J., Hajiyev, A., Nickel, S., Gen, M., Eds.; Springer: Singapore, 2017; pp. 1117–1134.

21. Hashim, M.; Nazam, M.; Abrar, M.; Hussain, Z.; Nazim, M.; Shabbir, R. Unlocking the Sustainable Production Indicators: A Novel TESCO Based Fuzzy AHP Approach. Cogent Bus. Manag. 2021, 8, 1870807, doi:10.1080/23311975.2020.1870807.

22. Kazibudzki, P.T.; Grzybowski, A.Z. On Some Advancements within Certain Multicriteria Decision Making Support Methodology. Am. J. Bus. Manag. 2013, 2, 143–154, doi:10.11634/216796061706281.

23. Kazibudzki, P.T. On the Statistical Discrepancy and Affinity of Priority Vector Heuristics in Pairwise-Comparison-Based Methods. Entropy 2021, 23, 1150, doi:10.3390/e23091150.

---

## [Editor Report · Decision Letter 2]

9 Aug 2023

PONE-D-23-12219R2On Uncertainty Related to the Inexactitude of Prioritization Based on Consistent Pairwise ComparisonsPLOS ONE

Dear Dr. Kazibudzki,

Thank you for submitting your manuscript to PLOS ONE. After careful consideration, we feel that it has merit but does not fully meet PLOS ONE’s publication criteria as it currently stands. Therefore, we invite you to submit a revised version of the manuscript that addresses the points raised during the review process.

We look forward to receiving your revised manuscript.

Kind regards,

Muhammad Hashim, PhD

Academic Editor

PLOS ONE

Journal Requirements:

Additional Editor Comments:

I appreciate the effort made by the author and consider some comments in revision. So, please incorporate following rest of the comments.

1. The title of study may be like that

‘The uncertainty in decision making process related to the inexactitude of prioritization based on consistent pairwise comparisons”

“The uncertainty related to the inexactitude of prioritization based on consistent pairwise comparisons”

The title start with the word “on” may not making a good sense. It is not compulsory for you to choose the propose one. You can also propose as you think suitable.

2. What is new in your study?” or statements like “Monte Carlo Simulation is an old fashioned approach/tool

In answer you can discuss your contributions even though you are using Monte Carlo Simulation but your answer is not showing good professionalism.

Dear Author, thanks for sharing the Journal policy link (https://journals.plos.org/plosone/s/editorial-and-peer-review-process) now I will request you please read the policy very carefully especially the headings “Editor Decision (The editor considers reviewer feedback and their own evaluation of the manuscript in order to reach a decision)”

3. Review of literature is not relevant to the study. Authors must add relevant studies of fuzzy AHP and traditional AHP method as well as Monte Carlo Simulation algorithm

For example, line No. 188, “the Weighted Product Model (WPM), see e.g. [120, 121]” but I did’t find the name of this method (WPM) in mentioned reference 120.

Line No. 186: problems: The Weighted Sum Model (WSM), see e.g. [3,118,119], but I did’t find the name of this method (WSM) in mentioned reference 03, Please recheck and answer this point.

Furthermore, I agreed with author the present moment contains 232 references, so author may reduce this number by deleting old references and focus on more relevant and updated references.

4. I appreciate the author for proofreading the paper but still there are some sentences that need to revise carefully.

Line No, 104-107: Taking into account the AHP drawbacks, many indicators of PCM consistency, commonly known as consistency indices (CIs), have been also proposed thus far, see e.g. [23,24,41,51–68] or, for example [69–77]. They also, due to brevity of this article, will not be scrutinized herein.

The sentences should be in a follow. Here are some examples. Please read carefully and improve the write up

The purpose of comments was to improve the research paper not confuse the author. The author is a direct beneficiary of a good publication and will have a good impact in his profile. So. Please take it positive and revise the paper as per comments.

---

## [Author Response · Author response to Decision Letter 2]

9 Aug 2023

Responses to Reviewers and the Academic Editor

The content of this document will be devoted exclusively to suggestions made by the editor as the manuscript at its current stage is fully and unconditionally accepted by two independent reviewers. They both claim that the manuscript is suitable for publication as it stands –

Reviewer #1: All comments have been addressed; The authors revised the manuscript according to the wise suggestion so the revised manuscript is suitable for publication.

Reviewer #2: All comments have been addressed. 

The author appreciates comments of the editor perfecting the manuscript and humbly introduce the editor’s suggestions everywhere they lead to the betterment of the research paper.

Additional Editor Comments:

I appreciate the effort made by the author and consider some comments in revision. So, please incorporate following rest of the comments.

1. The title of study may be like that ‘The uncertainty in decision making process related to the inexactitude of prioritization based on consistent pairwise comparisons”, “The uncertainty related to the inexactitude of prioritization based on consistent pairwise comparisons”

The title start with the word “on” may not making a good sense. It is not compulsory for you to choose the propose one. You can also propose as you think suitable.

The author appreciates the editor’s suggestion. The title was positively evaluated by the professional proofreader who is also the American native speaker. American English is not the author’s first language. The author depends on opinions of professionals. Hence, to acknowledge the editor’s viewpoint and appreciate the editor’s input to the problem, the author decided to exchange the word “on” on the word “The”.

2. What is new in your study?” or statements like “Monte Carlo Simulation is an old fashioned approach/tool. In answer you can discuss your contributions even though you are using Monte Carlo Simulation but your answer is not showing good professionalism. Dear Author, thanks for sharing the Journal policy link

(https://journals.plos.org/plosone/s/editorial-and-peer-review-process) now I will request you please read the policy very carefully especially the headings “Editor Decision (The editor considers reviewer feedback and their own evaluation of the manuscript in order to reach a decision)”.

The author would like to request the editor to understand that the written language connotations can sometimes be very confusing and sometimes it is very hard to understand the character of the writer’s intentions, especially when the stake is high and professionalism is interwoven with emotions like in this case. The author is fully aware that the “last” word in the case “to publish” or “not to publish” belongs to the editor and the editor’s professional judgment. However, the author would very much appreciate if the editor could choose to notice that the methodological contribution to the state of knowledge related to Monte Carlo Simulations was already indicated in the manuscript in its first and second revisions, please see e.g.

“As it seems the third issue is probably the most important problem in the contemporary arena of the MADM theory concerning AHP, and the only way to examine that phenomena is through computer simulations. It is the fact that Monte Carlo simulations are commonly recognized and applied as important and credible source of scientific information [1,2]. Their applications spread for examination purposes of various phenomena, e.g.: consequences of decisions made, or different processes subdued to random impact of the particular environment [3–10].”, 

and/or for example:

“However, only few research studies examine this problem from the perspective of PCM applicability for credible designation of decision maker’s (DM) priorities in the way that leads to minimization of the prioritization uncertainty related to possible, and sometimes very probable, ranking fluctuations. This problem constitutes the primary area of interest for this research paper as no research study was thus far identified that examines this problem via complex simulations from the perspective of consistent PCMs. So far, this concept has been studied only from the perspective of inconsistent PCMs, see e.g. [5,7,11–15]. As problems related to heuristics can be analyzed only via a computer simulation process, because they cannot be mathematically determined, the problem examined in this research paper is examined via Monte Carlo simulations, appropriately coded and executed with the application of Wolfram’s Mathematica Software.

It is the fact that Monte Carlo simulations are commonly recognized and applied as important and credible source of scientific information [1,2]. Their applications spread for examination purposes of various phenomena, e.g.: consequences of decisions made, or different processes subdued to random impact of the particular environment [3–10]. Hence, for the analysis of the problem revealed in the former subsections, the following Monte Carlo simulation algorithm is proposed (Fig 1-2). The primary version of the algorithm was devised and successfully applied for the first time by Grzybowski [14], and since then it was adapted and successfully implemented for many similar problems, see e.g. [5,7,11–13,15–18]. This research study applies the adaptation of this algorithm for examination of consistent PCMs.”.

…, and the fact that the editor indicated that the Monte Carlo Simulation methodology is not new but even an old fashioned concept, was understood by the author as the editor’s claim that the research paper lacks of its novelty to meet the journal’s standards – that is the only reason that these standards were brought to the editor’s attention. The author believes that now the editor will fully understand and appreciate the author’s intentions as fully professional.

3. Review of literature is not relevant to the study. Authors must add relevant studies of fuzzy AHP and traditional AHP method as well as Monte Carlo Simulation algorithm. For example, line No. 188, “the Weighted Product Model (WPM), see e.g. [120, 121]” but I did’t find the name of this method (WPM) in mentioned reference 120.

Line No. 186: problems: The Weighted Sum Model (WSM), see e.g. [3,118,119], but I did’t find the name of this method (WSM) in mentioned reference 03, Please recheck and answer this point.

Furthermore, I agreed with author the present moment contains 232 references, so author may reduce this number by deleting old references and focus on more relevant and updated references.

All the relevant studies of fuzzy AHP and traditional AHP method as well as Monte Carlo Simulation algorithm were already added to the manuscript during its first and second revisions. The editor’s suggestion to eliminate from the manuscript indicated references i.e. – Roy B. Decision-aid and decision-making. Eur J Oper Res. 1990; 45: 324–331. doi:10.1016/0377-2217(90)90196-I – former No. 3, and – Velasquez M, Hester P. An analysis of multi-criteria decision making methods. Int J Oper Res. 2013;10: 56–66 – former No. 120, was fully applied. The author would like his manuscript to be fully supported by related references. Hence, the author would like to kindly request the editor to appreciate the present value of the manuscript’s literature as it also may conduce to its readability and number of its quotations.

4. I appreciate the author for proofreading the paper but still there are some sentences that need to revise carefully. Line No, 104-107: Taking into account the AHP drawbacks, many indicators of PCM consistency, commonly known as consistency indices (CIs), have been also proposed thus far, see e.g. [23,24,41,51–68] or, for example [69–77]. They also, due to brevity of this article, will not be scrutinized herein. The sentences should be in a follow. Here are some examples. Please read carefully and improve the write up.

The author appreciates the editor’s suggestion. The questionable sentence was rephrased. American English is not the author’s first language. The author depends on opinions of professionals. The author can only hope that the editor will choose to appreciate the author’s efforts to meet the editor’s requirements.

The author deeply appreciates suggestions of the editor and believes that all improvements introduced to the manuscript as suggested by the editor will be also appreciated by the editor and will meet the editor’s expectations what will entail the unconditional acceptance of the manuscript for publication at its current stage in the PLoS ONE journal. 

The author can only believe that the editor will choose to take into account also the fact that the manuscript is already unconditionally approved for publication in the journal by two independent reviewers. 

Respectfully yours, 

Pawel Tadeusz Kazibudzki, PhD

References

1. Carmone, F.J.; Kara, A.; Zanakis, S.H. A Monte Carlo Investigation of Incomplete Pairwise Comparison Matrices in AHP. Eur. J. Oper. Res. 1997, 102, 538–553, doi:10.1016/S0377-2217(96)00250-0.

2. Herman, M.W.; Koczkodaj, W.W. A Monte Carlo Study of Pairwise Comparison. Inf. Process. Lett. 57, 25–29.

3. Kazibudzki, P.T. The Quality of Ranking during Simulated Pairwise Judgments for Examined Approximation Procedures. Model. Simul. Eng. 2019, 2019, e1683143, doi:10.1155/2019/1683143.

4. Zahedi, F. A Simulation Study of Estimation Methods in the Analytic Hierarchy Process. Socioecon. Plann. Sci. 1986, 20, 347–354, doi:10.1016/0038-0121(86)90046-7.

5. Kazibudzki, P.T. An Examination of Ranking Quality for Simulated Pairwise Judgments in Relation to Performance of the Selected Consistency Measure. Adv. Oper. Res. 2019, 2019, e3574263, doi:10.1155/2019/3574263.

6. Kazibudzki, P.T. An Examination of Performance Relations among Selected Consistency Measures for Simulated Pairwise Judgments. Ann. Oper. Res. 2016, 244, 525–544, doi:10.1007/s10479-016-2131-6.

7. Grzybowski, A.Z.; Starczewski, T. Simulation Analysis of Prioritization Errors in the AHP and Their Relationship with an Adopted Judgement Scale. In Proceedings of the Proceedings; San Francisco, USA, October 23 2018; p. 5.

8. Wu, H.; Leung, S.-O. Can Likert Scales Be Treated as Interval Scales?—A Simulation Study. J. Soc. Serv. Res. 2017, 43, 527–532, doi:10.1080/01488376.2017.1329775.

9. Zanakis, S.H.; Solomon, A.; Wishart, N.; Dublish, S. Multi-Attribute Decision Making: A Simulation Comparison of Select Methods. Eur. J. Oper. Res. 1998, 107, 507–529, doi:10.1016/S0377-2217(97)00147-1.

10. Winsberg, E. Science in the Age of Computer Simulation; University of Chicago Press: Chicago, IL, 2010; ISBN 978-0-226-90204-3.

11. Kazibudzki, P.T. On Estimation of Priority Vectors Derived from Inconsistent Pairwise Comparison Matrices. J. Appl. Math. Comput. Mech. 2022, 21, 52–59, doi:10.17512/jamcm.2022.4.05.

12. Kazibudzki, P.T.; Křupka, J. Pairwise Judgments Consistency Impact on Quality of Multi-Criteria Group Decision-Making with AHP. EM Ekon. Manag. 2019, 22, 195–212, doi:10.15240/tul/001/2019-4-013.

13. Kazibudzki, P.T. Redefinition of Triad’s Inconsistency and Its Impact on the Consistency Measurement of Pairwise Comparison Matrix. J. Appl. Math. Comput. Mech. 2016, 15, 71–78, doi:10.17512/jamcm.2016.1.07.

14. Grzybowski, A.Z. New Results on Inconsistency Indices and Their Relationship with the Quality of Priority Vector Estimation. Expert Syst. Appl. 2016, 43, 197–212, doi:10.1016/j.eswa.2015.08.049.

15. Grzybowski, A.Z.; Starczewski, T. New Look at the Inconsistency Analysis in the Pairwise-Comparisons-Based Prioritization Problems. Expert Syst. Appl. 2020, 113549, doi:10.1016/j.eswa.2020.113549.

16. Kazibudzki, P.T. The AHP Phenomenon of Rank Reversal Demystified.; December 2022.

17. Grzybowski, A.Z.; Starczewski, T. Remarks about Inconsistency Analysis in the Pairwise Comparison Technique. In Proceedings of the 2017 IEEE 14th International Scientific Conference on Informatics; November 2017; pp. 227–231.

18. Grzybowski, A.Z. On Some Recent Advancements within the Pairwise Comparison Methodology. In Proceedings of the 2017 IEEE 14th International Scientific Conference on Informatics; November 2017; pp. 1–5.

---

## [Editor Report · Decision Letter 3]

10 Aug 2023

PONE-D-23-12219R3The Uncertainty Related to the Inexactitude of Prioritization Based on Consistent Pairwise ComparisonsPLOS ONE

Dear Dr. Kazibudzki,

Thank you for submitting your manuscript to PLOS ONE. After careful consideration, we feel that it has merit but does not fully meet PLOS ONE’s publication criteria as it currently stands. Therefore, we invite you to submit a revised version of the manuscript that addresses the points raised during the review process.

We look forward to receiving your revised manuscript.

Kind regards,

Muhammad Hashim, PhD

Academic Editor

PLOS ONE

Journal Requirements:

Additional Editor Comments:

I appreciate the effort made by the author and consider comments in revision. So, please incorporate following rest of the comments.

1. The title of study may be like that

‘The uncertainty in decision making process related to the inexactitude of prioritization based on consistent pairwise comparisons”

“The uncertainty related to the inexactitude of prioritization based on consistent pairwise comparisons”

The title start with the word “on” may not making a good sense. It is not compulsory for you to choose the propose one. You can also propose as you think suitable.

2. What is new in your study?” or statements like “Monte Carlo Simulation is an old fashioned approach/tool

In answer you can discuss your contributions even though you are using Monte Carlo Simulation but your answer is not showing good professionalism.

Dear Author, thanks for sharing the Journal policy link (https://journals.plos.org/plosone/s/editorial-and-peer-review-process) now I will request you please read the policy very carefully especially the headings “Editor Decision (The editor considers reviewer feedback and their own evaluation of the manuscript in order to reach a decision)”

3. Review of literature is not relevant to the study. Authors must add relevant studies of fuzzy AHP and traditional AHP method as well as Monte Carlo Simulation algorithm

For example, line No. 188, “the Weighted Product Model (WPM), see e.g. [120, 121]” but I did’t find the name of this method (WPM) in mentioned reference 120.

Line No. 186: problems: The Weighted Sum Model (WSM), see e.g. [3,118,119], but I did’t find the name of this method (WSM) in mentioned reference 03, Please recheck and answer this point.

Furthermore, I agreed with author the present moment contains 232 references, so author may reduce this number by deleting old references and focus on more relevant and updated references.

4. I appreciate the author for proofreading the paper but still there are some sentences that need to revise carefully.

Line No, 104-107: Taking into account the AHP drawbacks, many indicators of PCM consistency, commonly known as consistency indices (CIs), have been also proposed thus far, see e.g. [23,24,41,51–68] or, for example [69–77]. They also, due to brevity of this article, will not be scrutinized herein.

The sentences should be in a follow. Here are some examples. Please read carefully and improve the write up

The purpose of comments was to improve the research paper not confuse the author. The author is a direct beneficiary of a good publication and will have a good impact in his profile. So. Please take it positive and revise the paper as per comments.

---

## [Author Response · Author response to Decision Letter 3]

10 Aug 2023

Responses to Reviewers and the Academic Editor

The content of this document will be devoted exclusively to suggestions made by the editor as the manuscript at its current stage is fully and unconditionally accepted by two independent reviewers. They both claim that the manuscript is suitable for publication as it stands –

Reviewer #1: All comments have been addressed; The authors revised the manuscript according to the wise suggestion so the revised manuscript is suitable for publication.

Reviewer #2: All comments have been addressed. 

The author appreciates comments of the editor perfecting the manuscript and humbly introduce the editor’s suggestions everywhere they lead to the betterment of the research paper.

Additional Editor Comments:

I appreciate the effort made by the author and consider some comments in revision. So, please incorporate following rest of the comments.

1. The title of study may be like that ‘The uncertainty in decision making process related to the inexactitude of prioritization based on consistent pairwise comparisons”, “The uncertainty related to the inexactitude of prioritization based on consistent pairwise comparisons”

The title start with the word “on” may not making a good sense. It is not compulsory for you to choose the propose one. You can also propose as you think suitable.

The author appreciates the editor’s suggestion. The title was positively evaluated by the professional proofreader who is also the American native speaker. American English is not the author’s first language. The author depends on opinions of professionals. Hence, to acknowledge the editor’s viewpoint and appreciate the editor’s input to the problem, the author decided to exchange the word “on” on the word “The”.

2. What is new in your study?” or statements like “Monte Carlo Simulation is an old fashioned approach/tool. In answer you can discuss your contributions even though you are using Monte Carlo Simulation but your answer is not showing good professionalism. Dear Author, thanks for sharing the Journal policy link

(https://journals.plos.org/plosone/s/editorial-and-peer-review-process) now I will request you please read the policy very carefully especially the headings “Editor Decision (The editor considers reviewer feedback and their own evaluation of the manuscript in order to reach a decision)”.

The author would like to request the editor to understand that the written language connotations can sometimes be very confusing and sometimes it is very hard to understand the character of the writer’s intentions, especially when the stake is high and professionalism is interwoven with emotions like in this case. The author is fully aware that the “last” word in the case “to publish” or “not to publish” belongs to the editor and the editor’s professional judgment. However, the author would very much appreciate if the editor could choose to notice that the methodological contribution to the state of knowledge related to Monte Carlo Simulations was already indicated in the manuscript in its first and second revisions, please see e.g.

“As it seems the third issue is probably the most important problem in the contemporary arena of the MADM theory concerning AHP, and the only way to examine that phenomena is through computer simulations. It is the fact that Monte Carlo simulations are commonly recognized and applied as important and credible source of scientific information [1,2]. Their applications spread for examination purposes of various phenomena, e.g.: consequences of decisions made, or different processes subdued to random impact of the particular environment [3–10].”, 

and/or for example:

“However, only few research studies examine this problem from the perspective of PCM applicability for credible designation of decision maker’s (DM) priorities in the way that leads to minimization of the prioritization uncertainty related to possible, and sometimes very probable, ranking fluctuations. This problem constitutes the primary area of interest for this research paper as no research study was thus far identified that examines this problem via complex simulations from the perspective of consistent PCMs. So far, this concept has been studied only from the perspective of inconsistent PCMs, see e.g. [5,7,11–15]. As problems related to heuristics can be analyzed only via a computer simulation process, because they cannot be mathematically determined, the problem examined in this research paper is examined via Monte Carlo simulations, appropriately coded and executed with the application of Wolfram’s Mathematica Software.

It is the fact that Monte Carlo simulations are commonly recognized and applied as important and credible source of scientific information [1,2]. Their applications spread for examination purposes of various phenomena, e.g.: consequences of decisions made, or different processes subdued to random impact of the particular environment [3–10]. Hence, for the analysis of the problem revealed in the former subsections, the following Monte Carlo simulation algorithm is proposed (Fig 1-2). The primary version of the algorithm was devised and successfully applied for the first time by Grzybowski [14], and since then it was adapted and successfully implemented for many similar problems, see e.g. [5,7,11–13,15–18]. This research study applies the adaptation of this algorithm for examination of consistent PCMs.”.

…, and the fact that the editor indicated that the Monte Carlo Simulation methodology is not new but even an old fashioned concept, was understood by the author as the editor’s claim that the research paper lacks of its novelty to meet the journal’s standards – that is the only reason that these standards were brought to the editor’s attention. The author believes that now the editor will fully understand and appreciate the author’s intentions as fully professional.

3. Review of literature is not relevant to the study. Authors must add relevant studies of fuzzy AHP and traditional AHP method as well as Monte Carlo Simulation algorithm. For example, line No. 188, “the Weighted Product Model (WPM), see e.g. [120, 121]” but I did’t find the name of this method (WPM) in mentioned reference 120.

Line No. 186: problems: The Weighted Sum Model (WSM), see e.g. [3,118,119], but I did’t find the name of this method (WSM) in mentioned reference 03, Please recheck and answer this point.

Furthermore, I agreed with author the present moment contains 232 references, so author may reduce this number by deleting old references and focus on more relevant and updated references.

All the relevant studies of fuzzy AHP and traditional AHP method as well as Monte Carlo Simulation algorithm were already added to the manuscript during its first and second revisions. The editor’s suggestion to eliminate indicated references i.e. – Roy B. Decision-aid and decision-making. Eur J Oper Res. 1990; 45: 324–331. doi:10.1016/0377-2217(90)90196-I, and – Velasquez M, Hester P. An analysis of multi-criteria decision making methods. Int J Oper Res. 2013;10: 56–66 – from the indicated positions, was fully applied. However, the author would like his manuscript to be fully supported by related and ample references. Hence, the author would like to kindly request the editor to appreciate the present value of the manuscript’s ample literature as it also may conduce to its readability and number of its quotations.

4. I appreciate the author for proofreading the paper but still there are some sentences that need to revise carefully. Line No, 104-107: Taking into account the AHP drawbacks, many indicators of PCM consistency, commonly known as consistency indices (CIs), have been also proposed thus far, see e.g. [23,24,41,51–68] or, for example [69–77]. They also, due to brevity of this article, will not be scrutinized herein. The sentences should be in a follow. Here are some examples. Please read carefully and improve the write up.

The author appreciates the editor’s suggestion. The questionable sentence was rephrased. American English is not the author’s first language. The author depends on opinions of professionals. The author can only hope that the editor will choose to appreciate the author’s efforts to meet the editor’s requirements.

The author deeply appreciates suggestions of the editor and believes that all improvements introduced to the manuscript as suggested by the editor will be also appreciated by the editor and will meet the editor’s expectations what will entail the unconditional acceptance of the manuscript for publication at its current stage in the PLoS ONE journal. 

The author can only believe that the editor will choose to take into account also the fact that the manuscript is already unconditionally approved for publication in the journal by two independent reviewers. 

Respectfully yours, 

Pawel Tadeusz Kazibudzki, PhD

References

1. Carmone, F.J.; Kara, A.; Zanakis, S.H. A Monte Carlo Investigation of Incomplete Pairwise Comparison Matrices in AHP. Eur. J. Oper. Res. 1997, 102, 538–553, doi:10.1016/S0377-2217(96)00250-0.

2. Herman, M.W.; Koczkodaj, W.W. A Monte Carlo Study of Pairwise Comparison. Inf. Process. Lett. 57, 25–29.

3. Kazibudzki, P.T. The Quality of Ranking during Simulated Pairwise Judgments for Examined Approximation Procedures. Model. Simul. Eng. 2019, 2019, e1683143, doi:10.1155/2019/1683143.

4. Zahedi, F. A Simulation Study of Estimation Methods in the Analytic Hierarchy Process. Socioecon. Plann. Sci. 1986, 20, 347–354, doi:10.1016/0038-0121(86)90046-7.

5. Kazibudzki, P.T. An Examination of Ranking Quality for Simulated Pairwise Judgments in Relation to Performance of the Selected Consistency Measure. Adv. Oper. Res. 2019, 2019, e3574263, doi:10.1155/2019/3574263.

6. Kazibudzki, P.T. An Examination of Performance Relations among Selected Consistency Measures for Simulated Pairwise Judgments. Ann. Oper. Res. 2016, 244, 525–544, doi:10.1007/s10479-016-2131-6.

7. Grzybowski, A.Z.; Starczewski, T. Simulation Analysis of Prioritization Errors in the AHP and Their Relationship with an Adopted Judgement Scale. In Proceedings of the Proceedings; San Francisco, USA, October 23 2018; p. 5.

8. Wu, H.; Leung, S.-O. Can Likert Scales Be Treated as Interval Scales?—A Simulation Study. J. Soc. Serv. Res. 2017, 43, 527–532, doi:10.1080/01488376.2017.1329775.

9. Zanakis, S.H.; Solomon, A.; Wishart, N.; Dublish, S. Multi-Attribute Decision Making: A Simulation Comparison of Select Methods. Eur. J. Oper. Res. 1998, 107, 507–529, doi:10.1016/S0377-2217(97)00147-1.

10. Winsberg, E. Science in the Age of Computer Simulation; University of Chicago Press: Chicago, IL, 2010; ISBN 978-0-226-90204-3.

11. Kazibudzki, P.T. On Estimation of Priority Vectors Derived from Inconsistent Pairwise Comparison Matrices. J. Appl. Math. Comput. Mech. 2022, 21, 52–59, doi:10.17512/jamcm.2022.4.05.

12. Kazibudzki, P.T.; Křupka, J. Pairwise Judgments Consistency Impact on Quality of Multi-Criteria Group Decision-Making with AHP. EM Ekon. Manag. 2019, 22, 195–212, doi:10.15240/tul/001/2019-4-013.

13. Kazibudzki, P.T. Redefinition of Triad’s Inconsistency and Its Impact on the Consistency Measurement of Pairwise Comparison Matrix. J. Appl. Math. Comput. Mech. 2016, 15, 71–78, doi:10.17512/jamcm.2016.1.07.

14. Grzybowski, A.Z. New Results on Inconsistency Indices and Their Relationship with the Quality of Priority Vector Estimation. Expert Syst. Appl. 2016, 43, 197–212, doi:10.1016/j.eswa.2015.08.049.

15. Grzybowski, A.Z.; Starczewski, T. New Look at the Inconsistency Analysis in the Pairwise-Comparisons-Based Prioritization Problems. Expert Syst. Appl. 2020, 113549, doi:10.1016/j.eswa.2020.113549.

16. Kazibudzki, P.T. The AHP Phenomenon of Rank Reversal Demystified.; December 2022.

17. Grzybowski, A.Z.; Starczewski, T. Remarks about Inconsistency Analysis in the Pairwise Comparison Technique. In Proceedings of the 2017 IEEE 14th International Scientific Conference on Informatics; November 2017; pp. 227–231.

18. Grzybowski, A.Z. On Some Recent Advancements within the Pairwise Comparison Methodology. In Proceedings of the 2017 IEEE 14th International Scientific Conference on Informatics; November 2017; pp. 1–5.

---

## [Editor Report · Decision Letter 4]

15 Aug 2023

The Uncertainty Related to the Inexactitude of Prioritization Based on Consistent Pairwise Comparisons

PONE-D-23-12219R4

Dear Dr. Kazibudzki,

We’re pleased to inform you that your manuscript has been judged scientifically suitable for publication and will be formally accepted for publication once it meets all outstanding technical requirements.

Kind regards,

Muhammad Hashim, PhD

Academic Editor

PLOS ONE
---

## [Editor Report · Acceptance letter]

29 Aug 2023

PONE-D-23-12219R4 

The uncertainty related to the inexactitude of prioritization based on consistent pairwise comparisons 

Dear Dr. Kazibudzki:

I'm pleased to inform you that your manuscript has been deemed suitable for publication in PLOS ONE. Congratulations! Your manuscript is now with our production department. 

Kind regards, 

on behalf of

Dr. Muhammad Hashim 

Academic Editor

PLOS ONE